# 💡 LaMP: Language-Motion Pretraining for Motion Generation, Retrieval, and Captioning

**Zhe Li**[1,2*]**, Weihao Yuan**[2*]**, Yisheng He**[2]**, Lingteng Qiu**[2]**, Shenhao Zhu**[2,3]**, Xiaodong Gu**[2]**,
Weichao Shen**[2]**, Yuan Dong**[2]**, Zilong Dong**[2†]**, Laurence T. Yang**[1†]

[1] Huazhong University of Science and Technology
[2] Alibaba Group
[3] Nanjing University

## Abstract

Language plays a vital role in the realm of human motion. Existing methods have largely depended on CLIP text embeddings for motion generation, yet they fall short in effectively aligning language and motion due to CLIP's pretraining on static image-text pairs. This work introduces **LaMP**, a novel **La**nguage-**M**otion **P**retraining model, which transitions from a language-vision to a more suitable language-motion latent space. It addresses key limitations by generating motion-informative text embeddings, significantly enhancing the relevance and semantics of generated motion sequences. With LaMP, we advance three key tasks: text-to-motion generation, motion-text retrieval, and motion captioning through aligned language-motion representation learning. For generation, we utilize LaMP to provide the text condition instead of CLIP, and an autoregressive masked prediction is designed to achieve mask modeling without rank collapse in transformers. For retrieval, motion features from LaMP's motion transformer interact with query tokens to retrieve text features from the text transformer, and vice versa. For captioning, we finetune a large language model with the language-informative motion features to develop a strong motion captioning model. In addition, we introduce the LaMP-BertScore metric to assess the alignment of generated motions with textual descriptions. Extensive experimental results on multiple datasets demonstrate substantial improvements over previous methods across all three tasks.

## 1 Introduction

Human motion represents an expanding frontier in computer vision, with substantial implications for diverse applications including film production, gaming, virtual reality, and robotics. Traditionally, textual representations have been favored for their simplicity and accessibility, enabling prominent tasks such as text-based motion retrieval, motion-to-text captioning, and text-driven motion generation. Despite the prevalent use of text in human motion tasks, challenges persist in achieving effective alignment between language and motion representations.

Previous methods usually leverage the pretrained CLIP (Radford et al., 2021) text encoder to extract embeddings from textual instructions for text-driven motion generation. Specifically, in many diffusion-based frameworks (Chen et al., 2023; Tevet et al., 2023; Lou et al., 2023; Zhang et al., 2023b), CLIP text embeddings are utilized as conditional signals to control the diffusion process. On the other hand, some transformer-based frameworks (Guo et al., 2022a; Zhang et al., 2023a; Zhong et al., 2023; Guo et al., 2023) stack the text embeddings with the input to guide the motion generation. In these frameworks, the extracted text embeddings serve as rich semantic representations and guide the generation process to produce motion sequences following the text instructions. In this paper, we seek to push the limit of the second path by designing a novel approach.

Although recent methods have achieved impressive results (Guo et al., 2023; Zhang et al., 2023a;b; Chen et al., 2023), they share a common drawback: relying on text embeddings from CLIP as condition signals. CLIP is well pretrained to align language and visual representation in the latent space. However, it is not optimal for aligning language and motion in two aspects. First, CLIP is pretrained using text-image pairs, which may capture text embeddings that primarily represent

---

*Equal Contribution     † Corresponding Author
Project Page: `https://aigc3d.github.io/LaMP/`

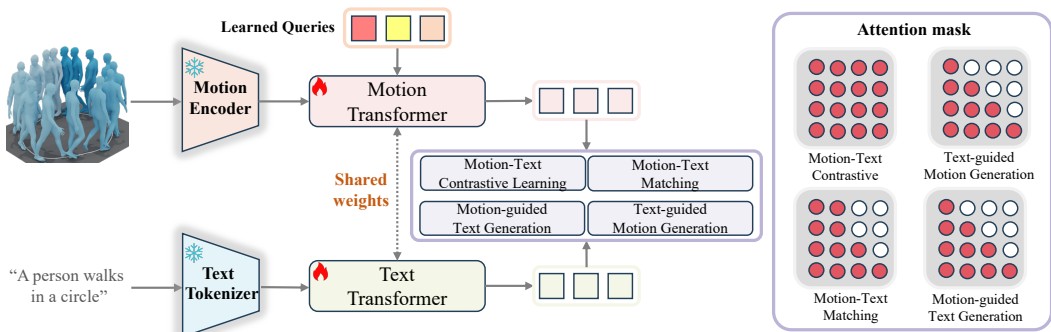

Figure 1: LaMP overview. We conduct joint training for contrastive learning, matching, and bidirectional text-motion translation by leveraging the textual features extracted from tokenized text descriptions via the text transformer and the motion features derived from the motion transformer.

image features, potentially overlooking information relevant to motion. Second, since CLIP is pretrained on static images, it focuses predominantly on static characteristics, disregarding the dynamic elements associated with motion. As a result, the text embeddings generated by CLIP may not sufficiently reflect the relationship between text and motion, leading to suboptimal performance in applications that demand a high degree of synchronization between verbal descriptions and corresponding movements.

To address this problem, we propose **LaMP**, achieving a paradigm shift from a language-vision latent space to a more appropriate language-motion latent space. This brings three benefits. First, the text embeddings directly model the dynamics in the motion rather than only modeling the static state. Second, instead of vision-informative, it can obtain motion-informative text embeddings as condition signals, which enhances the quality of generated motion, allowing for more contextually relevant and semantically aligned motion sequences. Third, along with the motion-informative text features, we can also acquire language-informative motion features. Thanks to these motion features, we achieve better motion-text retrieval and finetune a large language model (LLM) to perform motion captioning.

In this paper, starting from the language-motion pretraining model LaMP, we improve three human motion tasks, i.e. text-motion retrieval, text-to-motion generation, and motion-to-text captioning. For the LaMP pretraining, we perform language-motion representation learning by constraining the LaMP to derive text representations that are most pertinent to the corresponding motion. To achieve effective language-motion alignments, inspired by BLIP2 (Li et al., 2023) for language-vision alignment, four tasks are employed: motion-text contrastive learning, motion-text matching, motion-grounded text generation, and text-grounded motion generation. After pretraining, we utilize the extracted features **LaMP-Feat** for motion-text retrieval. The motion features from LaMP's motion transformer interact with query tokens to retrieve text features from the text transformer, and vice versa. Next, we propose a text-to-motion generation model **LaMP-T2M**. We adopt LaMP's text transformer in place of CLIP as the text encoder to extract text embeddings, which serve as the conditional signal to guide motion generation. Following previous works (Devlin, 2018; Guo et al., 2023; Pinyoanuntapong et al., 2024), we also employ the masked prediction technique. However, unlike previous methods, we adopt a decoder-only architecture, which alleviates the degradation in expressive power caused by low-rank matrices (Dong et al., 2021) during the training process to some extent. Additionally, the causal attention mask enhances information interaction within the masked regions. Subsequently, we propose a motion-to-text captioning model **LaMP-M2T**. We leverage the motion transformer from LaMP to obtain language-informative motion features, which are then fed into a LLM. By finetuning this LLM using LoRA (Hu et al., 2021), we develop a LLM capable of motion captioning. At last, we propose to evaluate the quality of generated motion using **LaMP-BertScore**. Specifically, we input the generated motion into LaMP-M2T to obtain the corresponding textual description, and then calculate the BertScore (Zhang et al., 2019) between the generated text and the ground truth text. This metric serves to assess how well the generated motions align with the true semantics.

Extensive experiments conducted across various datasets demonstrate the effectiveness of our approach in text-to-motion generation, motion-text retrieval, and motion-to-text captioning, with significant improvements compared to previous state-of-the-art methods. For instance, the FID is decreased by 28.9% on the HumanML3D dataset, and 28.0% on the KIT-ML dataset for motion generation. The primary contributions of this work are then summarized as follows:

- We novelly propose a **La**nguage-**M**otion **P**retraining model, termed as **LaMP**, and apply it to extract textual embeddings as conditional signals to guide motion generation. It not only ensures that the generated motions align more closely with semantic information but also reduces the gap between modalities. Furthermore, we observe that LaMP demonstrates outstanding capabilities in motion-text retrieval as well. To the best of our knowledge, we are the first to replace the language-vision space of CLIP with a language-motion space in motion generation.

- We propose a motion generation model LaMP-T2M, where we employ an autoregressive masked prediction mechanism. This approach alleviates the reduction in expressive capability caused by low-rank matrices during training and enhances the information interaction among masked regions.

- We propose a motion captioning model LaMP-M2T, where an LLM is fine-tuned from the LaMP motion feature. Based on this, we also introduce a new metric LaMP-BertScore for evaluating the extent to which generated motions align with semantic information.

- We outperform previous methods in three tasks including motion-text retrieval, motion-to-text captioning, and text-to-motion generation.

## 2 RELATED WORK

**Text-guided Human Motion Generation** Early research on text-to-motion generation can be divided into two primary approaches: diffusion-based continuous regression (Tevet et al., 2023; Chen et al., 2023; Zhang et al., 2022; Lou et al., 2023; Zhang et al., 2023b; Yuan et al., 2023) and transformer-based discrete classification (Zhang et al., 2023a; Zhong et al., 2023; Guo et al., 2023; Zou et al., 2024; Yuan et al., 2024). Among the diffusion-based methods, MotionDiffuse (Zhang et al., 2022) stands out as the first model to use fine-grained text instructions for body part motion. MDM (Tevet et al., 2023) learns the relationship between motion and input conditions on raw data, while MLD (Chen et al., 2023) reduces computational overhead with a latent space diffusion process. In the transformer-based category, motion is encoded with VQ-VAE (Van Den Oord et al., 2017) to create discrete tokens for prediction tasks. T2M-GPT (Zhang et al., 2023a) generates motion sequences from discretized inputs via a transformer. MoMask (Guo et al., 2022a) employs a motion residual VQ-VAE with multiple codebooks, enhancing generation through a residual transformer. This paper follows the transformer-based path but simplifies the process by utilizing a single codebook without a refinement step, resulting in a simpler and more flexible framework.

**Motion-Text Retrieval** Motion retrieval remains an under-explored area, which presents challenges due to its cross-modal nature, necessitating nearest-neighbor searches between text and motion modalities. Recent work T2M (Guo et al., 2022a) focuses on constructing a retrieval model for evaluation, using a margin-based contrastive loss and Euclidean distance for batch pairs. TEMOS (Petrovich et al., 2022) has a synthesis branch to generate motions from text and creates a cross-modal embedding space but only aligns positive pairs. TMR (Petrovich et al., 2023) builds upon TEMOS, and integrates a contrastive training strategy that incorporates negative examples, thus improving retrieval from a diverse set of fine-grained motions. In contrast, we adopt a completely different pipeline, using the motion and text transformer from pretrained LaMP directly for retrieval.

**Human Motion Captioning** To articulate human motion via natural language, (Takano & Nakamura, 2015) formulates a mapping between motion sequences and linguistic descriptions utilizing two statistical models. TM2T (Guo et al., 2022b) proposes a motion representation that condenses continuous motions into a concise sequence of discrete variables, leveraging a neural translation network for cross-modal mapping. Motiongpt (Jiang et al., 2023) introduces a motion-language training scheme with instruction tuning, to learn from task feedback and generate motion captions through prompts. DLP (Cai et al., 2024) constructs a new dataset MoCap and finetune a LLM on this dataset for motion captioning. Different from them, we propose a simpler approach, which utilizes language-informative motion text features and finetunes an LLM without instruction tuning.

## 3 METHOD

### 3.1 LaMP: LANGUAGE MOTION PRETRAINING

We present LaMP, a novel model that aligns motion and language more accurately than CLIP (Radford et al., 2021). Current approaches (Zhang et al., 2023a; Chen et al., 2023; Guo et al., 2023) leverage a

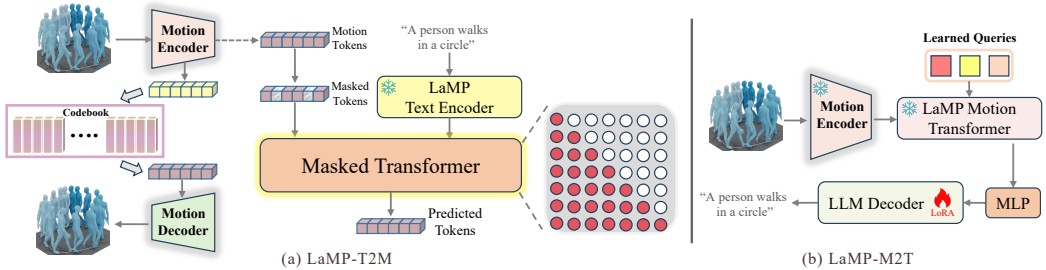

Figure 2: LaMP-T2M and LaMP-M2T frameworks overview. (**Left**) Pretrained LaMP's text transformer is employed to extract condition embedding and autoregressive mask prediction is performed. (**Right**) Finetuning an LLM to achieve motion captioning.

CLIP pretrained text encoder to extract text features as conditions for motion generation. However, the CLIP text encoder maps text features into an image-text aligned latent space, which may not perfectly align with motion features due to the domain gap between images and motions. Inspired by (Li et al., 2023), we develop a model that can better capture motion-informative text features and provide more precise conditions for motion generation. Furthermore, we leverage LaMP and learnable query tokens to project motion into the motion-language latent space. With this representation, we finetune a large language model and obtain a motion-to-text model.

### 3.1.1 PRELIMINARY

We adopt vanilla VQ-VAE (Van Den Oord et al., 2017) to convert a motion sequence into one tuple of discrete tokens. Specifically, we use 1D convolutional encoder $\mathcal{E}$ to encode the motion sequence $\mathcal{M}_{1:N} \in \mathbb{R}^{N \times D}$ to the latent vectors $m_{1:N} \in \mathbb{R}^{n \times d}$ with downsampling ratio of $N/n$ and latent dimension $d$. To obtain discrete tokens, we pre-define a learnable codebook $\mathcal{C} := \{(s, z_s)\}_{s \in S}^d$ for the latent vectors $m_{1:n}$, where $S$ is the size of codebook and $s$ is the index of embedding in $\mathcal{C}$. Each latent vector is replaced by the nearest codebook embedding in $\mathcal{C}$ according to the Euclidean distance, which can be formally denoted as: $\mathcal{Q}(m_i) = m_i \mapsto z_s$. The quantized code sequence $z_{1:n}$ is then projected back into the motion space to reconstruct the motion $\hat{m}$ via decoder $\mathcal{D}$.

### 3.1.2 MODEL ARCHITECTURE

Our aim is to align language and motion better. Similar to (Li et al., 2023), LaMP comprises two transformers (Vaswani, 2017) sub-modules that share the same self-attention layers: (1) a motion transformer that interacts with the frozen motion encoder and query tokens, and (2) a text transformer, as in Figure 1. The motion transformer learns to extract a set of salient motion features from the input motion, while the text transformer learns to generate a corresponding set of motion-informative textual outputs. By jointly optimizing these two sub-modules with four auxiliary tasks, LaMP can effectively extract motion-informative text features and language-informative motion features.

We define a set of learnable query tokens $q$, which serve as the input to the motion transformer. These queries interact with each other through self-attention layers and with motion features via cross-attention layers. Moreover, the queries can engage with the text through the same self-attention mechanism. Following (Li et al., 2023), we initialize LaMP with the pre-trained BERT_base (Devlin, 2018), with the cross-attention layers randomly initialized.

### 3.1.3 MOTION-LANGUAGE ALIGNMENT

We jointly optimize four distinct objectives that share a common input format and model parameters. Each objective employs a unique attention masking strategy to modulate the interaction between motion and language.

**Motion-Text Contrastive Learning**    Contrastive learning is a key technique for aligning modalities. Similar to CLIP (Radford et al., 2021), we perform motion-text contrastive learning in LaMP. It aims to bring positive samples closer together and push negative samples further apart in the latent space, thereby maximizing the mutual information between motion and language. We first input the motion embeddings output from the motion encoder into the motion transformer and interact with the

query tokens $q$, obtaining the motion feature $f_m$, and input the text into the text transformer to obtain the text feature $f_t$. Then the pairwise similarity is computed between $f_m$ and $f_t$, and we select the highest one as the motion-text similarity. Following (Li et al., 2023), to avoid information leakage, we employ a unimodal self-attention mask, where motions and texts are not allowed to see each other.

**Motion-Text Matching** Matching tasks can enable finer-grained alignment between modalities. We frame the task as a binary classification, where the model is required to predict whether a given motion-text pair is matched or unmatched. We employ a bi-directional self-attention mechanism, enabling the model to capture the interdependent relationships between motion and language. The motion embeddings, text, and query tokens are fed together into the transformer, and the output is passed to a binary linear classifier to obtain logits, which are then averaged across the batch to obtain the matching score. The matching task further increases the mutual information between motion and language, and achieves information interaction.

**Motion-grounded Text Generation** To further enhance the alignment between the two modalities and get the informative features, we hope to achieve cross-modal translation in LaMP. Given a motion sequence as the condition, we intend to generate the corresponding text. We first utilize pretrained tokenizer to tokenize the texts and obtain ground truth labels. Then we input the motion embedding and query token $q$ into the motion transformer to obtain the motion feature $f_m$, which is then fed into the text transformer. To generate texts autoregressively, we employ a causal attention mask and compute the classification loss $\mathcal{L}_{mgt}$ on the output tokens.

**Text-grounded Motion Generation** Different from BLIP-2 (Li et al., 2023), we perform text-grounded motion generation in LaMP, thanks to pretrained motion VQ-VAE in stage 1, to enhance the alignment. BLIP-2 (Li et al., 2023) primarily focuses on extracting visual features that can be comprehended by LLMs. However, we require not only language-informative motion features but also motion-informative text features. Consequently, we incorporate text-grounded motion generation during the pretraining phase of LaMP. The ground truth label of motion sequence can be obtained from the pretrained codebook $\mathcal{C} \in \mathbb{R}^{S \times d}$ in the first stage: $\{b_i = \mathcal{Q}(m_i)\}_{i=1\ldots n}$. We input the text into the text transformer and interact with query tokens $q$ via the motion transformer's cross-attention layers. The resulting features are then used for autoregressive generation through the motion transformer. Additionally, we define a motion classifier head $F$ to map the generated results $\{g_i\}_{i=1\ldots n}$ to the space $\in \mathbb{R}^S$, and compute the classification loss with the ground truth label.

$$\mathcal{L}_{tgm} = \sum_{i=1}^{n} \text{Cross-Entropy}(F(g_i), b_i) \tag{1}$$

The joint training across the four aforementioned tasks endows LaMP with strong motion-language alignment capability. After training, the text transformer of LaMP can extract motion-informative text features, making it a better condition encoder than CLIP for motion generation (Section 3.2). Given the current R-Precision Top-1 metric only achieves around 52% accuracy on real data, LaMP can also serve as a better motion-text retrieval model (Section 3.3) and an evaluator for the R-Precision and Multimodal Distance metrics, as verified in the experiments. Additionally, the motion transformer can also extract language-informative motion features. We utilize language-informative motion features based on pretrained LLM to train a motion-to-text model, and based on this model, we propose using the LaMP-BertScore metric to evaluate the quality of generated motion (Section 3.4).

### 3.2 LaMP-T2M: Motion Generation from Text

Drawing inspiration from (Guo et al., 2023), a mask transformer is employed in our work to generate motion tokens. As depicted in Figure 2, we first randomly replace a subset of the sequence elements $\{m_1, m_2, ...., m_n\}$ with a special $[M]$ token. Let $m^M = \{m_1, m_2, ..., [M], m_{n-2}, [M], m_n\}$ denote the sequence after this masking process. Our aim is to accurately predict the masked tokens, given the context text $t$ and the partially masked sequence $m^M$, thereby endowing the model with generative capabilities. Different from existing approaches, we use pretrained LaMP for extracting text features. The optimization objective is to minimize the negative log-likelihood of prediction:

$$\mathcal{L}_{mask} = \sum_{m_k^M = [M]} - \log p(m_k^M | m^M, t) \tag{2}$$

### 3.2.1 MASK STRATEGY

We adopt the same mask strategy as MoMask (Guo et al., 2023). We employ a cosine function $\gamma$ to determine the masking ratio. Specifically, the mask ratio is calculated as $\gamma(r) = \cos(\pi r/2) \in [0, 1]$, where $r \in [0, 1]$ and $r = 0$ means a fully masked sequence. During the training process, we randomly sample $r$ from a uniform distribution $U(0, 1)$, and then we uniformly mask $\lceil \gamma(r) \cdot n \rceil$ tokens of the whole sequence.

Additionally, we also perform the remasking strategy utilized in the BERT (Devlin, 2018). Specifically, if a token is chosen for masking, we replace it with $[M]$ token with probability 80%, with a random token with probability 10%, and keep it unchanged with probability 10%.

### 3.2.2 AUTOREGRESSIVE GENERATION

Unlike the bidirectional attention mask in MoMask (Guo et al., 2023), we employ a causal attention mask for autoregressive mask prediction tasks. Currently, transformer-based motion generation models (Guo et al., 2023; Zhang et al., 2023a) commonly utilize bidirectional attention masks, which correspond to encoder-only model architectures. However, during training, the bidirectional attention mask allows the model to simultaneously rely on contextual information, simplifying the mask prediction task and diminishing the model's generative capacity.

In addition, this bidirectional masking leads to rank collapse. The attention matrix generated by a bidirectional attention mask typically arises from the product of a low-rank decomposed matrix and a softmax function; specifically, it results from multiplying an $n \times d$ matrix with a $d \times n$ matrix before applying softmax (where $n \gg d$). This form of attention matrix suffers from reduced expressiveness due to low-rank issues (Dong et al., 2021). In contrast, the attention matrix for a causal attention mask is a lower triangular matrix, with its determinant equal to the product of its diagonal elements. Due to the presence of softmax, all diagonal elements must be positive, ensuring that its determinant is also positive. Consequently, the attention matrix of the causal attention mask (decoder-only architecture) is guaranteed to be full-rank, providing greater expressiveness. Therefore, we predict the masked regions autoregressively:

$$\max_{\theta} \mathbb{E}\Big[ \sum_{i=1}^{n} \log P_\theta(m_i^M | t, m_{<i}^M) \Big] \tag{3}$$

### 3.3 LAMP-FEAT: MOTION TEXT RETRIEVAL

Given a text query $T$—for instance, "A person is walking in a circle"—our primary objective is to rank motions from a comprehensive database based on their semantic alignment with the textual input. The ultimate aim is to retrieve the motion that exhibits the highest correspondence with the provided textual description, effectively bridging the gap between disparate modalities—text and motion. We hypothesize that a model exhibiting better alignment between these modalities will inherently possess improved cross-modal retrieval capabilities, and can serve as a more reasonable R-Precision evaluator.

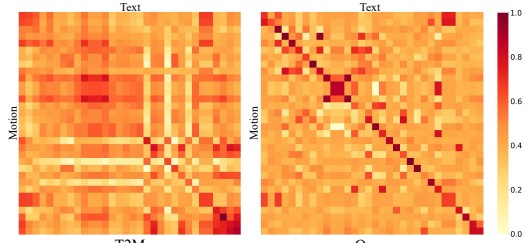

Figure 3: Heatmap of similarity matrix. The diagonal represents positive sample pairs, with darker colors indicating better quality.

To facilitate this alignment, we leverage the motion embeddings obtained from a frozen motion encoder, which undergoes processing through the motion transformer component of LaMP. This setup enables the embeddings to interact dynamically with the query tokens $q$, yielding a refined motion feature representation $f_m$. Such refinement is pivotal for accurately retrieving the corresponding text feature $f_t$, as generated by the text transformer. Conversely, our architecture is designed to support reciprocal interactions; the text embeddings can similarly inform and enhance the motion representations. This bidirectional exchange fosters a richer understanding of the underlying semantics, ultimately ensuring that the retrieved motion sequences are not merely similar, but truly relevant and contextually aligned with the input query. As shown in Figure 3, to validate that LaMP can better match motion-text pairs, we present the similarity matrix in the form of a heatmap. The results indicate that LaMP demonstrates more superior retrieval capabilities than T2M (Guo et al., 2022b).

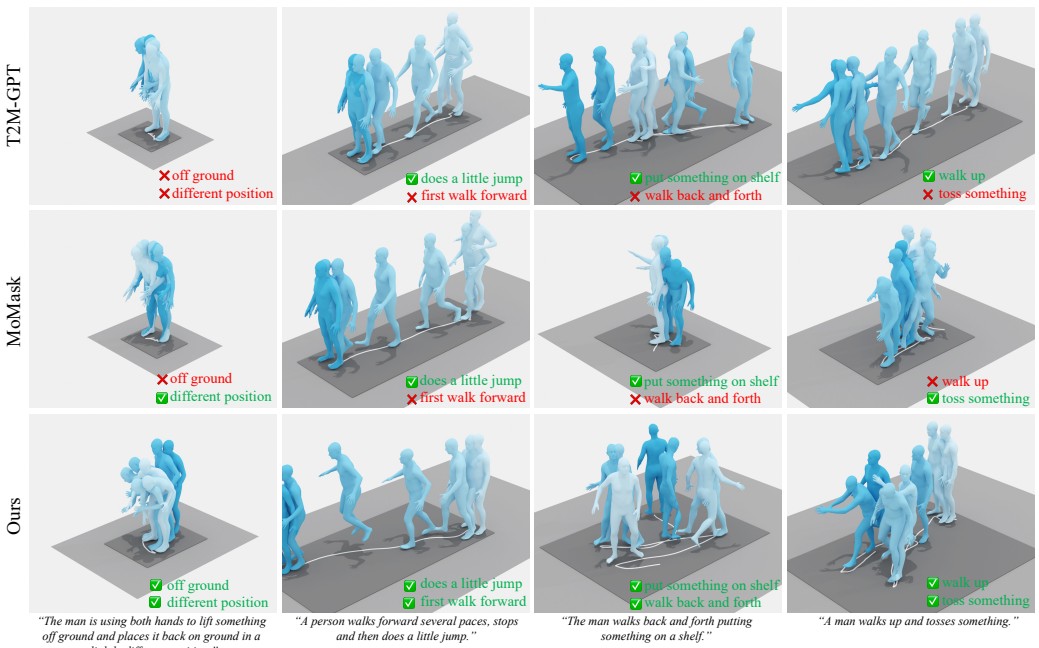

Figure 4: Qualitative results of text-to-motion generation on HumanML3D.

## 3.4 LaMP-M2T: Motion to Text Captioning

With the assistance of LaMP, we finetune a pretrained LLM (OPT-2.7b (Zhang et al., 2023c)) to generate corresponding text from motion using LoRA (Hu et al., 2021). As depicted in Figure 2, we employ a fully connected layer to linearly project the output motion feature $f_m$ into the same dimensional space as the text embeddings of the LLM. We input these language-informative motion features from LaMP and a prompt into LLM. Since LaMP has been pretrained to extract semantically meaningful motion representations, it effectively acts as an information bottleneck, passing the most useful information to the language model.

We input the generated motion into LaMP-M2T to obtain the textual description of the generated motion, and compute the BertScore against the ground truth text, termed as LaMP-BertScore, which serves to evaluate the extent to which the generated motion aligns with the semantic information.

## 3.5 Inference

During the inference process, we initiate with a fully masked sequence $m^M(0)$ and aim to generate the complete sequence over $K$ iterations. At the $k$-th iteration, given the partially masked token sequence $m^M(k)$, the model first estimates the probability distribution for the tokens at the masked positions and samples motion tokens based on this distribution. Subsequently, the tokens with the lowest confidence scores, specifically the lowest $\lceil \gamma(r) \cdot n \rceil$ values, are re-masked, while the remaining tokens stay unchanged for subsequent iterations. The updated token sequence $m^M(k+1)$ serves as the basis for predicting the token sequence in the following iteration until $k$ reaches $K$. Afterward, all tokens are decoded into motion with pretrained VQVAE decoder in stage 1.

**Classifier-free Guidance** The classifier-free guidance (CFG) method (Chang et al., 2023; Ho & Salimans, 2022) is employed to integrate text embeddings into the transformer framework. In the training stage, the transformer is trained unconditionally with a probability of 10%. During inference, CFG is implemented at the last linear projection layer right before the softmax operation. At this point, the final logits $l_f$, are computed by adjusting the conditional logits $l_c$ relative to the unconditional logits $l_{uc}$, using a guidance scale denoted as $\alpha$, here $\alpha$ is set as 4:

$$l_f = (1 + \alpha) \cdot l_c - \alpha \cdot l_{uc} \tag{4}$$

| Methods | R Precision↑ | | | FID↓ | LaMP-BertScore↑ | MultiModal Dist↓ | Diversity → |
|---|---|---|---|---|---|---|---|
| | Top 1 | Top 2 | Top 3 | | | | |
| Ground Truth | $0.511^{\pm.003}$ | $0.703^{\pm.003}$ | $0.797^{\pm.002}$ | $0.002^{\pm.00}$ | 100.00 | $2.974^{\pm.008}$ | $9.503^{\pm.065}$ |
| TM2T (Guo et al., 2022b) | $0.424^{\pm.003}$ | $0.618^{\pm.003}$ | $0.729^{\pm.002}$ | $1.501^{\pm.017}$ | - | $3.467^{\pm.011}$ | $8.589^{\pm.076}$ |
| T2M (Guo et al., 2022a) | $0.455^{\pm.003}$ | $0.636^{\pm.003}$ | $0.736^{\pm.002}$ | $1.087^{\pm.021}$ | - | $3.347^{\pm.008}$ | $9.175^{\pm.083}$ |
| MDM (Tevet et al., 2023) | - | - | $0.611^{\pm.007}$ | $0.544^{\pm.044}$ | - | $5.566^{\pm.027}$ | $\mathbf{9.559^{\pm.086}}$ |
| MLD (Chen et al., 2023) | $0.481^{\pm.003}$ | $0.673^{\pm.003}$ | $0.772^{\pm.002}$ | $0.473^{\pm.013}$ | 52.08 | $3.196^{\pm.010}$ | $9.724^{\pm.082}$ |
| MotionDiffuse (Zhang et al., 2022) | $0.491^{\pm.001}$ | $0.681^{\pm.001}$ | $0.782^{\pm.001}$ | $0.630^{\pm.001}$ | - | $3.113^{\pm.001}$ | $9.410^{\pm.049}$ |
| T2M-GPT (Zhang et al., 2023a) | $0.492^{\pm.003}$ | $0.679^{\pm.002}$ | $0.775^{\pm.002}$ | $0.141^{\pm.005}$ | 56.21 | $3.121^{\pm.009}$ | $9.761^{\pm.081}$ |
| PhysDiff (Yuan et al., 2023) | - | - | 0.631 | 0.433 | - | - | - |
| MotionGPT (Zhang et al., 2024) | - | - | - | 0.567 | - | 3.775 | - |
| M2DM (Kong et al., 2023) | $0.497^{\pm.003}$ | $0.682^{\pm.003}$ | $0.763^{\pm.002}$ | $0.352^{\pm.005}$ | - | $2.974^{\pm.016}$ | $9.926^{\pm.073}$ |
| Fg-T2M (Wang et al., 2023) | $0.492^{\pm.002}$ | $0.683^{\pm.003}$ | $0.783^{\pm.002}$ | $0.243^{\pm.019}$ | - | $3.109^{\pm.007}$ | $9.278^{\pm.072}$ |
| AttT2M (Zhong et al., 2023) | $0.499^{\pm.003}$ | $0.690^{\pm.002}$ | $0.786^{\pm.002}$ | $0.112^{\pm.006}$ | - | $3.038^{\pm.007}$ | $9.700^{\pm.090}$ |
| DiverseMotion (Lou et al., 2023) | $0.496^{\pm.004}$ | $0.687^{\pm.004}$ | $0.783^{\pm.003}$ | $0.070^{\pm.004}$ | - | $3.063^{\pm.011}$ | $9.551^{\pm.068}$ |
| ParCo (Zou et al., 2024) | $0.515^{\pm.003}$ | $0.706^{\pm.003}$ | $0.801^{\pm.002}$ | $0.109^{\pm.005}$ | - | $2.927^{\pm.008}$ | $9.576^{\pm.088}$ |
| MMM (Pinyoanuntapong et al., 2024) | $0.504^{\pm.003}$ | $0.696^{\pm.003}$ | $0.794^{\pm.002}$ | $0.080^{\pm.003}$ | - | $2.998^{\pm.007}$ | $9.411^{\pm.058}$ |
| ReMoDiffuse (Zhang et al., 2023b) | $0.510^{\pm.005}$ | $0.698^{\pm.004}$ | $0.795^{\pm.004}$ | $0.103^{\pm.004}$ | - | $2.974^{\pm.016}$ | $9.018^{\pm.075}$ |
| MoMask (Guo et al., 2023) | $0.521^{\pm.002}$ | $0.713^{\pm.002}$ | $0.807^{\pm.002}$ | $0.045^{\pm.002}$ | 60.40 | $2.958^{\pm.008}$ | - |
| Ours | $\mathbf{0.557^{\pm.003}}$ | $\mathbf{0.751^{\pm.002}}$ | $\mathbf{0.843^{\pm.001}}$ | $\mathbf{0.032^{\pm.002}}$ | 60.81 | $\mathbf{2.759^{\pm.007}}$ | $9.571^{\pm.069}$ |
| Ground Truth | $0.424^{\pm.005}$ | $0.649^{\pm.006}$ | $0.779^{\pm.006}$ | $0.031^{\pm.004}$ | 100.00 | $2.788^{\pm.012}$ | $11.080^{\pm.097}$ |
| TM2T (Guo et al., 2022b) | $0.280^{\pm.005}$ | $0.463^{\pm.006}$ | $0.587^{\pm.005}$ | $3.599^{\pm.153}$ | - | $4.591^{\pm.026}$ | $9.473^{\pm.0117}$ |
| T2M (Guo et al., 2022a) | $0.361^{\pm.005}$ | $0.559^{\pm.007}$ | $0.681^{\pm.007}$ | $3.022^{\pm.107}$ | - | $2.052^{\pm.107}$ | $10.72^{\pm.145}$ |
| MDM (Tevet et al., 2023) | - | - | $0.396^{\pm.004}$ | $0.497^{\pm.021}$ | - | $9.191^{\pm.022}$ | $10.85^{\pm.109}$ |
| MLD (Chen et al., 2023) | $0.390^{\pm.008}$ | $0.609^{\pm.008}$ | $0.734^{\pm.007}$ | $0.404^{\pm.027}$ | 48.47 | $3.204^{\pm.027}$ | $10.80^{\pm.117}$ |
| MotionDiffuse (Zhang et al., 2022) | $0.417^{\pm.004}$ | $0.621^{\pm.004}$ | $0.739^{\pm.004}$ | $1.954^{\pm.062}$ | - | $2.958^{\pm.005}$ | $\mathbf{11.10^{\pm.143}}$ |
| T2M-GPT (Zhang et al., 2023a) | $0.416^{\pm.006}$ | $0.627^{\pm.006}$ | $0.745^{\pm.006}$ | $0.514^{\pm.029}$ | 46.53 | $3.007^{\pm.023}$ | $10.86^{\pm.049}$ |
| PhysDiff (Yuan et al., 2023) | $0.510^{\pm.005}$ | $0.698^{\pm.006}$ | $0.795^{\pm.004}$ | $0.103^{\pm.004}$ | - | $2.974^{\pm.016}$ | - |
| MotionGPT (Zhang et al., 2024) | $0.510^{\pm.005}$ | $0.698^{\pm.006}$ | $0.795^{\pm.004}$ | $0.103^{\pm.004}$ | - | $2.974^{\pm.016}$ | 10.54 |
| M2DM (Kong et al., 2023) | $0.416^{\pm.004}$ | $0.628^{\pm.004}$ | $0.743^{\pm.004}$ | $0.515^{\pm.029}$ | - | $3.015^{\pm.017}$ | $11.417^{\pm.097}$ |
| Fg-T2M (Wang et al., 2023) | $0.418^{\pm.005}$ | $0.626^{\pm.004}$ | $0.745^{\pm.004}$ | $0.571^{\pm.047}$ | - | $3.114^{\pm.015}$ | $10.93^{\pm.083}$ |
| AttT2M (Zhong et al., 2023) | $0.413^{\pm.006}$ | $0.632^{\pm.006}$ | $0.751^{\pm.006}$ | $0.870^{\pm.039}$ | - | $3.039^{\pm.021}$ | $10.96^{\pm.123}$ |
| DiverseMotion (Lou et al., 2023) | $0.416^{\pm.005}$ | $0.637^{\pm.008}$ | $0.760^{\pm.011}$ | $0.468^{\pm.098}$ | - | $2.892^{\pm.041}$ | $10.873^{\pm.101}$ |
| ParCo (Zou et al., 2024) | $0.430^{\pm.004}$ | $0.649^{\pm.007}$ | $0.772^{\pm.006}$ | $0.453^{\pm.027}$ | - | $2.820^{\pm.028}$ | $10.95^{\pm.094}$ |
| MMM (Pinyoanuntapong et al., 2024) | $0.3/4^{\pm.005}$ | $0.590^{\pm.006}$ | $0.718^{\pm.005}$ | $0.429^{\pm.019}$ | - | $3.146^{\pm.019}$ | $10.633^{\pm.097}$ |
| ReMoDiffuse (Zhang et al., 2023b) | $0.427^{\pm.014}$ | $0.641^{\pm.004}$ | $0.765^{\pm.055}$ | $0.155^{\pm.006}$ | - | $2.814^{\pm.012}$ | $10.80^{\pm.105}$ |
| MoMask (Guo et al., 2023) | $0.433^{\pm.007}$ | $0.656^{\pm.005}$ | $0.781^{\pm.005}$ | $0.204^{\pm.011}$ | 56.89 | $2.779^{\pm.022}$ | - |
| Ours | $\mathbf{0.479^{\pm.006}}$ | $\mathbf{0.691^{\pm.005}}$ | $\mathbf{0.826^{\pm.005}}$ | $\mathbf{0.141^{\pm.013}}$ | 57.54 | $\mathbf{2.704^{\pm.018}}$ | $10.929^{\pm.101}$ |

Table 1: The quantitative results of text-to-motion generation with evaluator following previous methods on the HumanML3D dataset and the KIT-ML dataset.

| Datasets | Methods | LaMP-R Precision↑ | | | LaMP-R MultiModal Dist ↓ |
|---|---|---|---|---|---|
| | | Top 1 | Top 2 | Top 3 | |
| HumanML3D | T2M-GPT (Zhang et al., 2023a) | $0.796^{\pm.002}$ | $0.90.4^{\pm.003}$ | $0.92.4^{\pm.001}$ | $1.549^{\pm.008}$ |
| | MoMask (Guo et al., 2023) | $0.824^{\pm.002}$ | $0.937^{\pm.002}$ | $0.957^{\pm.001}$ | $1.306^{\pm.006}$ |
| | **Ours** | $\mathbf{0.867^{\pm.002}}$ | $\mathbf{0.962^{\pm.001}}$ | $\mathbf{0.981^{\pm.001}}$ | $\mathbf{1.187^{\pm.008}}$ |
| KIT-ML | T2M-GPT(Zhang et al., 2023a) | $0.710^{\pm.002}$ | $0.844^{\pm.002}$ | $0.896^{\pm.001}$ | $1.493^{\pm.019}$ |
| | MoMask (Guo et al., 2023) | $0.732^{\pm.002}$ | $0.871^{\pm.002}$ | $0.922^{\pm.001}$ | $1.289^{\pm.027}$ |
| | **Ours** | $\mathbf{0.784^{\pm.002}}$ | $\mathbf{0.907^{\pm.001}}$ | $\mathbf{0.963^{\pm.001}}$ | $\mathbf{1.174^{\pm.013}}$ |

Table 2: Evaluation results of text-to-motion generation with LaMP evaluator on T2M-GPT, MoMask, and ours.

## 4 EXPERIMENTS

### 4.1 EXPERIMENTS SETTINGS

**Datasets** We evaluate our model on HumanML3D (Guo et al., 2022a) and KIT-ML (Plappert et al., 2016) datasets. The HumanML3D dataset comprises 14,616 motion sequences accompanied by 44,970 textual descriptions, while the KIT-ML dataset includes 3,911 movement sequences and 6,278 text inputs. In line with the methodologies established in prior research (Guo et al., 2022a), we allocate 23,384 samples for training, 1,460 for validation, and 4,383 for testing within HumanML3D, and utilize 4,888 for training, 300 for validation, and 830 for testing in KIT-ML. The extracted motion poses yield motion features with dimensionalities of 263 for HumanML3D and 251 for KIT-ML. These motion features encapsulate both global attributes, such as root velocity, root height, and foot contact, as well as local details including joint positions, velocities, and rotations relative to the root.

| Methods | Text-Motion Retrieval↑ | | | | | Motion-Text Retrieval↑ | | | | |
| --- | --- | --- | --- | --- | --- | --- | --- | --- | --- | --- |
| | R@1↑ | R@2↑ | R@3↑ | R@5↑ | R@10↑ | R@1↑ | R@2↑ | R@3↑ | R@5↑ | R@10↑ |
| **HumanML3D** | | | | | | | | | | |
| TEMOS (Petrovich et al., 2022) | 0.424 | 53.52 | 61.14 | 70.96 | 84.15 | 39.96 | 53.49 | 61.79 | 72.40 | 85.89 |
| T2M (Guo et al., 2022a) | 52.48 | 71.05 | 80.65 | 89.66 | **96.58** | 52.00 | 71.21 | 81.11 | 89.87 | 96.78 |
| TMR (Petrovich et al., 2023) | 67.16 | 81.32 | 86.81 | 91.43 | 95.36 | 67.97 | 81.20 | 86.35 | 91.70 | 95.27 |
| Ours | $67.18^{\pm0.5}$ | $81.9^{\pm0.4}$ | $87.04^{\pm0.3}$ | $92.0^{\pm0.2}$ | $95.73^{\pm0.2}$ | $68.02^{\pm0.3}$ | $82.1^{\pm0.3}$ | $87.5^{\pm0.3}$ | $92.2^{\pm0.3}$ | $96.9^{\pm0.3}$ |
| **KIT-ML** | | | | | | | | | | |
| T2MOS (Petrovich et al., 2022) | 43.88 | 58.25 | 67.00 | 74.00 | 84.75 | 41.88 | 55.88 | 65.62 | 75.25 | 85.75 |
| T2M (Guo et al., 2022a) | 42.25 | 62.62 | 75.12 | 87.50 | 96.12 | 39.75 | 62.75 | 73.62 | 86.88 | 95.88 |
| TMR (Petrovich et al., 2023) | 49.25 | 69.75 | 78.25 | 87.88 | 95.00 | 50.12 | 67.12 | 76.88 | 88.88 | 94.75 |
| Ours | $52.5^{\pm0.7}$ | $74.8^{\pm0.5}$ | $84.7^{\pm0.5}$ | $92.7^{\pm0.3}$ | $97.6^{\pm0.3}$ | $54.0^{\pm.005}$ | $75.3^{\pm0.5}$ | $84.4^{\pm0.4}$ | $92.2^{\pm0.2}$ | $97.6^{\pm0.2}$ |

Table 3: Text-motion (**left**) and motion-text (**right**) retrieval benchmark on the HumanML3D and KIT-ML.

| Method | R Precision↑ | | MultiModal Dist↓ | LaMP-BertScore↑ | Bleu@1 ↑ | Bleu@4 ↑ | Rouge↑ | Cider↑ |
| --- | --- | --- | --- | --- | --- | --- | --- | --- |
| | Top 1 | Top 3 | | | | | | |
| T2MT (Guo et al., 2022a) | 0.516 | 0.823 | 2.935 | 32.2 | **48.9** | 7.00 | **38.1** | 16.8 |
| Motiongpt (Jiang et al., 2023) | 0.543 | 0.827 | 2.821 | 32.4 | 48.2 | 12.47 | 37.4 | **29.2** |
| LaMP-M2T (Ours) | **0.547** | **0.831** | **2.808** | **32.7** | 47.8 | **13.04** | 37.1 | 28.9 |

Table 4: The quantitative results of motion captioning on the HumanML3D, we adhere to the evaluation frameworks outlined in (Jiang et al., 2023).

The local joint data accounted for corresponds to 22 and 21 joints, respectively, from the SMPL model (Loper et al., 2023) for the HumanML3D and KIT-ML datasets.

**Implementation details** Our model is implemented on NVIDIA A100 GPU using PyTorch. For the motion VQ-VAE, we employ resblocks for both the encoder and decoder, with a downscale factor of 4. The VQ consists of 6 quantization layers, where each layer's codebook contains 512 512-dimensional codes. The quantization dropout ratio $p$ is set to 0.2. The masked transformer is composed of 6 transformer layers with casual attention masks, 6 heads, and a latent dimension of 384. The learning rate reaches 2e-4 after 2000 iterations with a linear warm-up schedule for the training of all models. During inference, we set the CFG scale of mask transformer as 4 on HumanML3D, and 2 on KIT-ML. Meanwhile, $K$ was set to 10 on both datasets.

## 4.2 EVALUATION

**Evaluation of Text-to-Motion Generation** We conduct an extensive evaluation of our model against prior text-to-motion approaches, encompassing both diffusion-based and transformer-based models. Our results, summarized in Table 1, reveal that our method demonstrably outperforms all previous methods on both the HumanML3D and KIT-ML datasets. Notably, our model achieves an improvement of 6.9%, 5.3%, and 4.7% in R-Precision Top {1, 2, 3} on HumanML3D, respectively. Furthermore, we also improve FID by 28.9% on HumanML3D, and achieve state-of-the-art LaMP-BertScore, highlighting the remarkable fidelity of our generated motions. Qualitative results presented in Figure 4 corroborate these findings, demonstrating that our method produces motions exhibiting a significantly better alignment with the input text compared to existing techniques.

We also report LaMP-R Precision and LaMP-Multimodal Distance obtained by extracting motion embeddings and text embeddings using LaMP in Table 2, showing significant improvements over the existing evaluator. This reveals that the LaMP evaluator could better evaluate methods in the motion generation task.

**Evaluation of Motion-Text Retrieval** To prove the retrieval ability of LaMP, we also evaluate standard retrieval performance for both text-to-motion and motion-to-text tasks, using metrics analogous to the R-Precision. These metrics include R@{1,2,3,5,10}, where a higher value signifies better performance. Recall at rank $k$ represents the proportion of instances where the correct label appears within the top $k$ retrieved results. Importantly, all retrieval evaluations are conducted on a test set of TMR (Petrovich et al., 2023) that includes unseen real motion. Our results including text-motion retrieval and motion-text retrieval, are summarized in Table 3.

**Evaluation of Motion-to-Text Captioning** Motion-to-text entails generating textual descriptions based on sequences of motion. We perform a comparative analysis of the LaMP-M2T against the recent TM2T (Guo et al., 2022a) and Motiongpt Jiang et al. (2023). Our evaluation is conducted on the HumanML3D dataset, utilizing the metrics proposed in (Guo et al., 2022a). In Table 4, we

| | R Precision↑ | | | FID↓ | LaMP-BertScore↑ | MultiModal Dist↓ |
|---|---|---|---|---|---|---|
| | Top 1 | Top 2 | Top 3 | | | |
| LaMP with contrastive learning | $0.519^{\pm.002}$ | $0.714^{\pm.005}$ | $0.811^{\pm.002}$ | $0.084^{\pm.002}$ | 57.61 | $2.876^{\pm.009}$ |
| + motion-text mathcing | $0.547^{\pm.002}$ | $0.732^{\pm.002}$ | $0.824^{\pm.002}$ | $0.067^{\pm.002}$ | 58.90 | $2.855^{\pm.007}$ |
| + motion-grounded text generation | $0.549^{\pm.002}$ | $0.737^{\pm.003}$ | $0.828^{\pm.002}$ | $0.062^{\pm.002}$ | 60.47 | $2.806^{\pm.007}$ |
| + text-grounded motion generation | $\mathbf{0.557^{\pm.003}}$ | $\mathbf{0.751^{\pm.002}}$ | $\mathbf{0.843^{\pm.001}}$ | $\mathbf{0.032^{\pm.002}}$ | **60.81** | $\mathbf{2.759^{\pm.007}}$ |

Table 5: Ablation study of the impact of different tasks in LaMP on generative performance on HumanML3D.

| | R Precision↑ | | | FID↓ | LaMP-BertScore↑ | MultiModal Dist↓ |
|---|---|---|---|---|---|---|
| | Top 1 | Top 2 | Top 3 | | | |
| Baseline with CLIP | $0.510^{\pm.002}$ | $0.701^{\pm.006}$ | $0.796^{\pm.002}$ | $0.086^{\pm.002}$ | 57.49 | $2.992^{\pm.009}$ |
| + LaMP | $0.548^{\pm.002}$ | $0.748^{\pm.002}$ | $0.840^{\pm.002}$ | $0.040^{\pm.002}$ | 59.84 | $2.794^{\pm.007}$ |
| + Query tokens | $0.550^{\pm.002}$ | $0.741^{\pm.002}$ | $0.834^{\pm.002}$ | $0.042^{\pm.002}$ | 60.54 | $2.9783^{\pm.006}$ |
| + Decoder-only | $\mathbf{0.557^{\pm.003}}$ | $\mathbf{0.751^{\pm.002}}$ | $\mathbf{0.843^{\pm.001}}$ | $\mathbf{0.032^{\pm.002}}$ | **60.81** | $\mathbf{2.759^{\pm.007}}$ |

Table 6: Ablation study of text-to-motion generation on HumanML3D. We report the impact of LaMP's text encoder, interactions with query tokens, and the mask prediction manner on the results.

follow the settings in  (Jiang et al., 2023) and adopt the raw ground truth text descriptions to facilitate a more precise evaluation. The results indicate that LaMP-M2T has competitive performance in generating text descriptions corresponding to motion sequences.

### 4.3 ABLATION STUDY

In the ablation experiments, we primarily examine the contributions of LaMP, query tokens, and autoregressive mask prediction to the quality of motion generation.

**Proxy Tasks in LaMP**   To validate the significance of each task during the LaMP training process, we report the impact of each task on the generated results in Table 5. The text-grounded motion generation task aids LaMP in extracting motion-informative text features, making this task the most influential on the generation results.

**LaMP as Text Encoder**   To validate the effectiveness of our proposed LaMP, we first establish a baseline model that utilizes CLIP as the text encoder. We then replace CLIP's text encoder with our proposed LaMP text transformer, using the resulting text features as new conditions to guide the motion generation. Based on the results presented in Table 6, we observe significant improvements in the model across all metrics.

**Interact with Query Tokens**   As shown in Figure 1, during the training process of LaMP, the query tokens $q$ do not directly interact with the text features. However, we believe that the query tokens carry a significant amount of motion information, which enhances the accuracy of the conditional guidance. Therefore, we further enable interaction between the text features and the query tokens through cross-attention layers, resulting in improved generative effects shown in Table 6.

**Decoder-only Mask Transformer**   Unlike MoMask (Guo et al., 2023), our model employs an autoregressive mask transformer. We believe that the bidirectional attention mask suffers from low-rank issues during training (Yang et al., 2024), which reduces the model's expressive ability and lacks information interaction between the masked regions. We adopt a decoder-only model architecture with a causal attention mask, which mitigates the low-rank problem of the matrix and enhances information interaction between the masked areas. As shown in Table 6, the decoder-only model architecture achieves better generative performance.

## 5 CONCLUSION

In this work, we introduce *LaMP*, a pioneering framework that bridges the gap between language and motion, achieving unprecedented alignment and performance in the process. Our findings elucidate how leveraging a nuanced approach to motion-text relationships can significantly enhance the generation of 3D human poses conditioned on textual descriptions. By integrating a decoder-only mask transformer, our *LaMP-T2M* achieves state-of-the-art results in human motion generation. Concurrently, we propose the *LaMP-M2T* model for action description and introduce the *LaMP-BertScore* metric to evaluate the quality of the generated motions. Comprehensive experiments validate the effectiveness of the proposed approach.

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
