# A APPENDIX

## A.1 INTRODUCTION

This is the supplementary material, which is divided into the following sections:

1. Metrics details are elaborated in Section A.2.

2. Implementation details are presented in Section A.3, including VQ-VAE implementation details A.3.1, LaMP implementation details A.3.2, and training and inference details for motion generation A.3.3.

3. More qualitative results are shown in Section A.8, including motion-to-text captioning and text-to-motion generation.

## A.2 METRICS

The evaluation metrics from (Guo et al., 2022a) employed in this study include: (1) *Frechet Inception Distance* (FID), which assesses the quality of generated motions by examining the discrepancies in the distribution of high-level features between synthetic and authentic motion samples; (2) *R-Precision* and *Multimodal Distance*, which measure the degree of semantic alignment between the provided textual input and the generated motions; (3) *Multimodality* that evaluates the diversity of motions produced from identical textual prompts, and (4) *BertScore*, which evaluates the quality of generated motion. We believe that motion should align with semantic information, rather than being fully consistent with the ground truth motion. We input the generated motion $\hat{m}$ into LaMP-M2T to obtain caption $\hat{t}$ and calculate the *BertScore* between the ground truth $t$ and $\hat{t}$ to assess the extent to which the generated motion conforms to the semantics.

## A.3 IMPLEMENTATION DETAILS

### A.3.1 VQ-VAE IMPLEMENTATION DETAILS

**Motion VQ-VAE** Continuous features often suffer from the problem of data sparsity, where most feature combinations are not present in the training set. Discretization can transform continuous features into a more limited set of categorical features, converting the generation problem into a classification task. Therefore, we first discretize the continuous motion features. A vanilla VQ-VAE (Van Den Oord et al., 2017) can be utilized to convert a motion sequence into one tuple of discrete tokens. We employ a straightforward convolutional architecture consisting of 1D convolutions, residual blocks, and ReLU activations. Figure A1 depicts the framework of VQVAE. Temporal downsampling is achieved via convolutions with a stride of 2, while nearest neighbor interpolation is utilized for upsampling.

Specifically, we use 1D convolutional encoder $\mathcal{E}$ encode the motion sequence $\mathcal{M}_{1:N} \in \mathbb{R}^{N \times D}$ to the latent vectors $m_{1:N} \in \mathbb{R}^{n \times d}$ with downsampling ratio of $N/n$ and latent dimension $d$. To obtain discrete tokens, we pre-define a learnable codebook $\mathcal{C} := \{(s, z_s)\}_{s \in S}^d$ for the latent vectors $m_{1:n}$, where $S$ is the size of codebook and $s$ is the index of embedding in $\mathcal{C}$.

Subsequently, as described in Eq. 5, each latent vector is calculated the distance with the embeddings in the codebook $\mathcal{C}$, and then replaced by the codebook embedding that has the nearest distance to the original latent vector, which can be formally denoted as: $\mathcal{Q}(m_i) = m_i \mapsto z_s$. The quantized code sequence $z_{1:n}$ is then projected back into the motion space to reconstruct the motion $\hat{m}$ vid decoder $\mathcal{D}$. After all, the indices of the selected embeddings in codebook $\mathcal{C}$, referred to as motion tokens, serve as an alternative discrete representation of the input motion.

$$\mathcal{Q}(m_i) = z_s, \text{where } s = \arg\min_{j \in \{1...S\}} \|z_j - m_i\|_2 \tag{5}$$

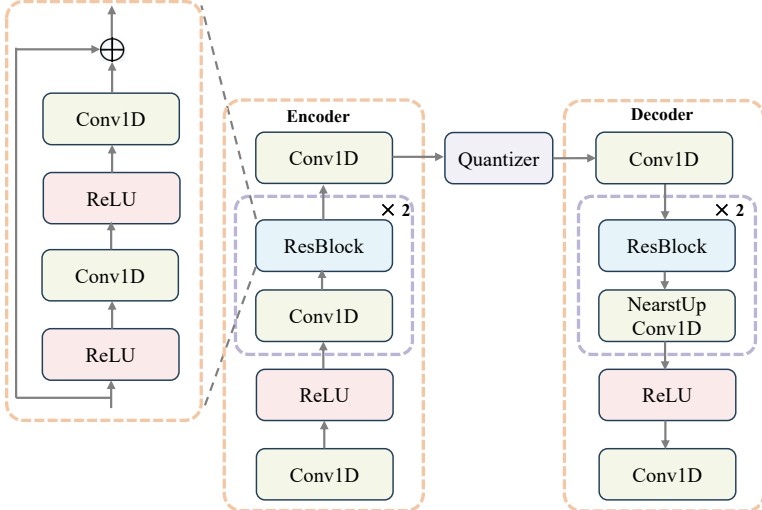

Figure A1: Overview of VQVAE.

**Optimization Goal** The optimization objective of VQ-VAE $\mathcal{L}_{vq}$ comprises three key components: the reconstruction loss $\mathcal{L}_{recon}$, the embedding loss $\mathcal{L}_{emb}$ and the commitment loss $\mathcal{L}_{com}$.

$$\mathcal{L}_{vq} = \mathcal{L}_{recon} + \underbrace{\sum_{i=0}^{n} \|\mathrm{sg}[m_i] - z_s\|_2}_{\mathcal{L}_{emb}} + \beta \underbrace{\sum_{i=0}^{n} \|m_i - \mathrm{sg}[z_s]\|_2}_{\mathcal{L}_{com}}, \tag{6}$$

where $\beta$ is a hyper-parameter for the commitment loss and sg denotes the stop-gradient. Following T2M-GPT (Zhang et al., 2023a), we employ L1 smooth loss and an additional regularization on the velocity. We use $m$ to denote the original motion and $\hat{m}$ to represent the reconstructed motion. $V(m) = \{v_i = m_{i+1} - m_i\}_{i=\{1...n\}}$ represents the veloctiy of motion sequence $m$. Therefore, the $\mathcal{L}_{recon}$ can be formulated as:

$$\mathcal{L}_{recon} = \|m - \hat{m}\|_1 + \alpha \|V(m) - V(\hat{m})\|_1, \tag{7}$$

where $\alpha$ is a hyper-parameter to balance the two losses.

**Optimization strategy** Since the codebook collapse situation (Razavi et al., 2019; Van Den Oord et al., 2017) can be found in the naive VQ-VAE, we perform exponential moving average (EMA) and codebook reset (Code Reset) to alleviate this problem. EMA facilitates the smooth evolution of the codebook $\mathcal{C}$: $\lambda \mathcal{C}^{t-1} + (1 - \lambda) \to \mathcal{C}^t$, where $\mathcal{C}^t$ represents the codebook at iteration $t$ and $\lambda$ denotes the exponential moving constant. The code Reset technique identifies inactive codes during the training process and subsequently reassigns them based on the input data.

### A.3.2 LaMP IMPLEMENTATION DETAILS

During the training of LaMP, we utilize the motion encoder pretrained in VQ-VAE and keep its weights frozen. Following BLIP-2 (Li et al., 2023), we initialize LaMP's text transformer and motion transformer with the pretrained weights of BERT$_{base}$, whereas the cross-attention layers in the motion transformer are randomly initialized. Depicted in Figure A2 (a), two transformers share the same self-attention layers, but the motion transformer has specialized cross-attention layers designed for interaction with query tokens. We set the sequence length of queries as 49, each with a dimensionality of 768, matching the hidden dimension of the motion transformer and text transformer. The query tokens pass through the self-attention layer of the motion transformer and interact with the motion through cross-attention layers.

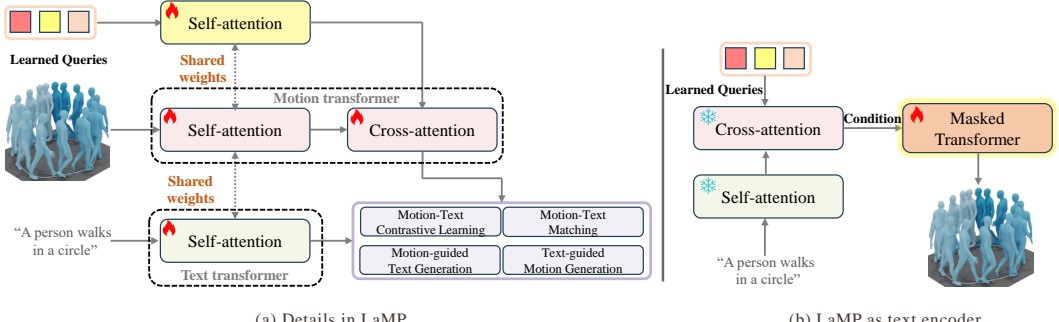

(a) Details in LaMP

(b) LaMP as text encoder

Figure A2: (**Left**) Details in LaMP. Motion transformer consists of self-attention layers and cross-attention layers (interact with query tokens), while text transformer only has self-attention layers. (**Right**) LaMP extracts text features as condition signals.

| Methods | R Precision↑ | | | FID↓ | MultiModal Dist↓ | Diversity → |
|---|---|---|---|---|---|---|
| | Top 1 | Top 2 | Top 3 | | | |
| HumanML3D-LaMP-T2M | $0.423^{\pm.006}$ | $0.657^{\pm.005}$ | $0.771^{\pm.005}$ | $0.226^{\pm.012}$ | $2.768^{\pm.022}$ | $10.536^{\pm.098}$ |
| Ours | $\mathbf{0.479^{\pm.006}}$ | $\mathbf{0.691^{\pm.005}}$ | $\mathbf{0.826^{\pm.005}}$ | $\mathbf{0.141^{\pm.013}}$ | $\mathbf{2.704^{\pm.018}}$ | $10.929^{\pm.101}$ |

Table A1: The quantitative results of text-to-motion generation on the KIT-ML dataset with LaMP pretrained on HumanML3D (the first row) and KIT-ML (the second row).

### A.3.3 TRAINING AND INFERENCE DETAILS FOR MOTION GENERATION

As illustrated in Figure A2 (b), we utilize LaMP as the text encoder, where the text passes through the self-attention layers of the text transformer. Subsequently, it interacts with the query tokens via the cross-attention layers of the motion transformer, serving as conditions to guide the motion generation.

During the training process, we employ a causal attention mask for autoregressive generation, which mitigates the effects of low-rank matrices and enhances information interaction among the masked regions. During the inference phase, we align our strategy with that of MoMask (Guo et al., 2023), utilizing a bi-directional attention mask.

### A.4 GENERALIZATION ABILITY OF LAMP

To validate the generalization ability of LaMP, we conduct additional experiments. Specifically, we utilize the LaMP model pretrained on the HumanML3D dataset as a text encoder and re-train the text-to-motion task on the KIT-ML dataset. The results are presented in Table A1. It can be observed that motion generation performance has a slight decline; however, it still maintains a commendable level, thereby demonstrating a degree of generalization. In future work, we will endeavor to build larger datasets and retrain LaMP, aiming to enhance its generalization ability.

### A.5 INFERENCE TIME

Following MoMask (Guo et al., 2023), we assess the efficiency and quality of motion generation compared to some methods in Figure A3. The inference cost is quantified as the average inference time across 100 samples executed on a single Nvidia 2080Ti device. Our findings indicate that LaMP achieves a more advantageous balance between generation quality and computational efficiency when compared to baseline methods. This may be attributed to MoMask's multi-layer Residual-VQ structure, along with the refinement of the inference process through the Residual Transformer, resulting in reduced time consumption on our part.

### A.6 ANALYSIS ON FAILURE CASES

We identify two primary categories of failure cases. The first category arises from issues inherent in the dataset's textual annotations. As depicted in Figure A4, the textual prompt is "A person raises his

left arm." However, the dataset itself contains incorrect motion that leads to failures in our model's outputs. Similarly, for the prompt "A person kicks with his right leg and then kicks with his left leg," the ground truth does not include the motion "kick with left leg," which results in failures in our generated results.

The second one is in the context of our experiments with cartoon characters, we encountered a challenge. Cartoon characters frequently perform motions that are impossible for humans, such as flying. As illustrated in Figure A5, when we use the text prompt "A person stands and flies up," the results does not meet our expectations. This shortfall is attributed to the current dataset's relatively limited size and lack of generalizability. Moving forward, it is crucial to integrate the motions of cartoon characters with those of humans to construct a more extensive dataset to address this issue comprehensively.

## A.7 DIFFERENCES FROM PREVIOUS METHODS

**Differences from MotionGPT and DLP on motion captioning.** MotionGPT (Zhang et al., 2024) requires instruction tuning of the LLM for the motion captioning task, our method for motion captioning does not necessitate such an operation, as our motion embeddings are inherently compatible with the LLM's understanding. Consequently, we have effectively addressed the intermodal gap DLP (Cai et al., 2024) also utilizes the instruction tuning technique to empower the LLM with motion captioning ability. Moreover, it constructs a new dataset, MoCap. Finetuning LLM on this dataset enhances motion captioning performance.

**Differences from HumanTomato on motion-language alignment.** Our method differs from HumanTomato (Lu et al., 2023) in terms of model architecture and training objectives. While HumanTomato employs the motion encoder and text encoder from TMR (Petrovich et al., 2023), LaMP utilizes a transformer-based motion encoder and text encoder similar to that of CLIP. Furthermore, our training objective is not limited to contrastive loss; we designed four proxy tasks including contrastive learning, matching tasks, motion-grounded text generation, and text-grounded motion generation. This design enhances the alignment effectiveness of LaMP. As a result, we can obtain motion-aware text embeddings as well as language-aware motion embeddings, making it easier for us to finetune a motion captioning LLM.

## A.8 MORE QUALITATIVE RESULTS OF MOTION-TO-TEXT CAPTIONING AND TEXT-TO-MOTION GENERATION

Qualitative results of motion-to-text captioning are presented in Figure A6, which demonstrate that our LaMP-M2T has the capability to generate textual descriptions of the motion sequence.

Moreover, we present additional qualitative results in Figures A7 and A8. As illustrated in Figure A8, several issues in MoMask, such as floating, are effectively alleviated in our approach. Moreover, our model demonstrates a higher sensitivity to numerical values; for instance, when instructed to "A man jumps twice in place.", our generated motion will accurately reflect this by jumping exactly twice, while MoMask results in more jumps. In contrast, T2M-GPT does not perform any jumps at all. Analysis of the generated motion sequences reveals that our method is capable of producing intricate and engaging motions that align well with the provided text prompts.

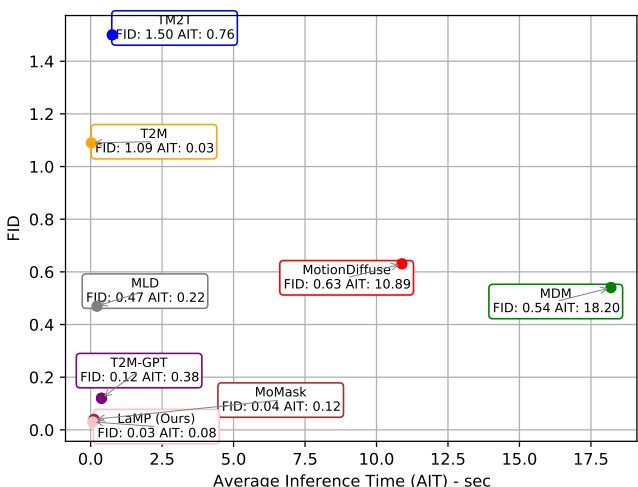

Figure A3: Comparisons on FID and Inference Cost.

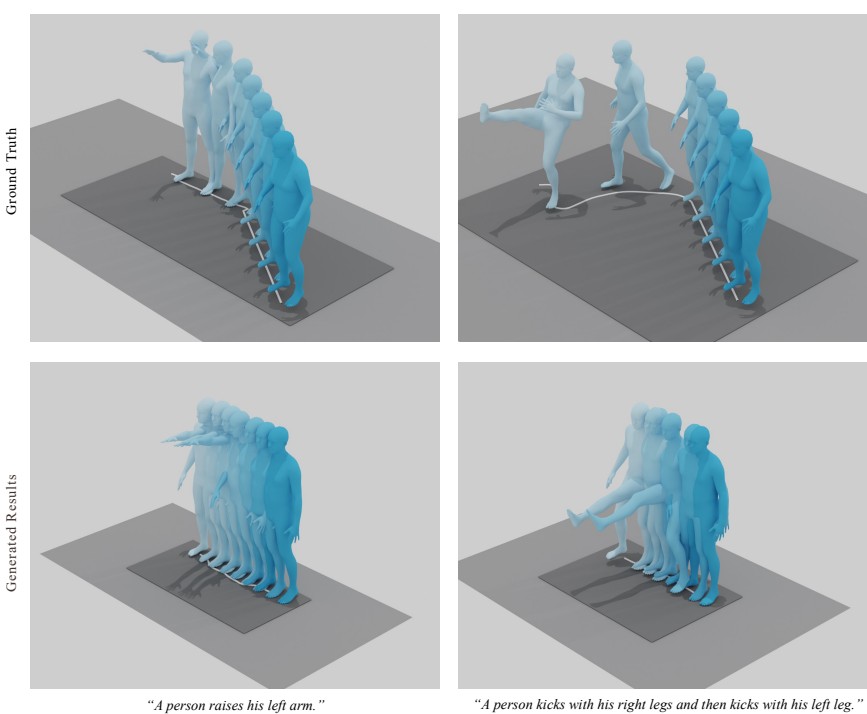

*"A person raises his left arm."*     *"A person kicks with his right legs and then kicks with his left leg."*

Figure A4: The failure cases due to the wrong textual annotations in the dataset.

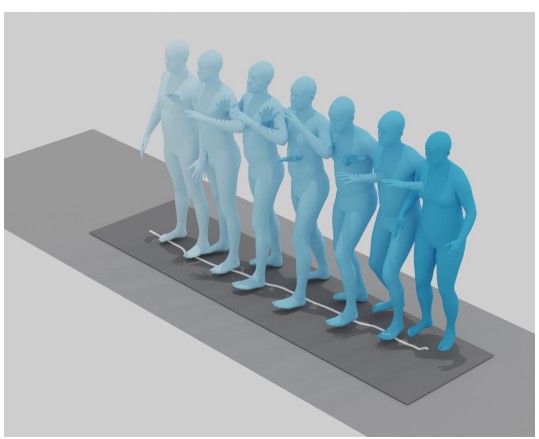

*"A person stands and then flies up."*

Figure A5: The failure cases due to the unseen motion and text prompt.

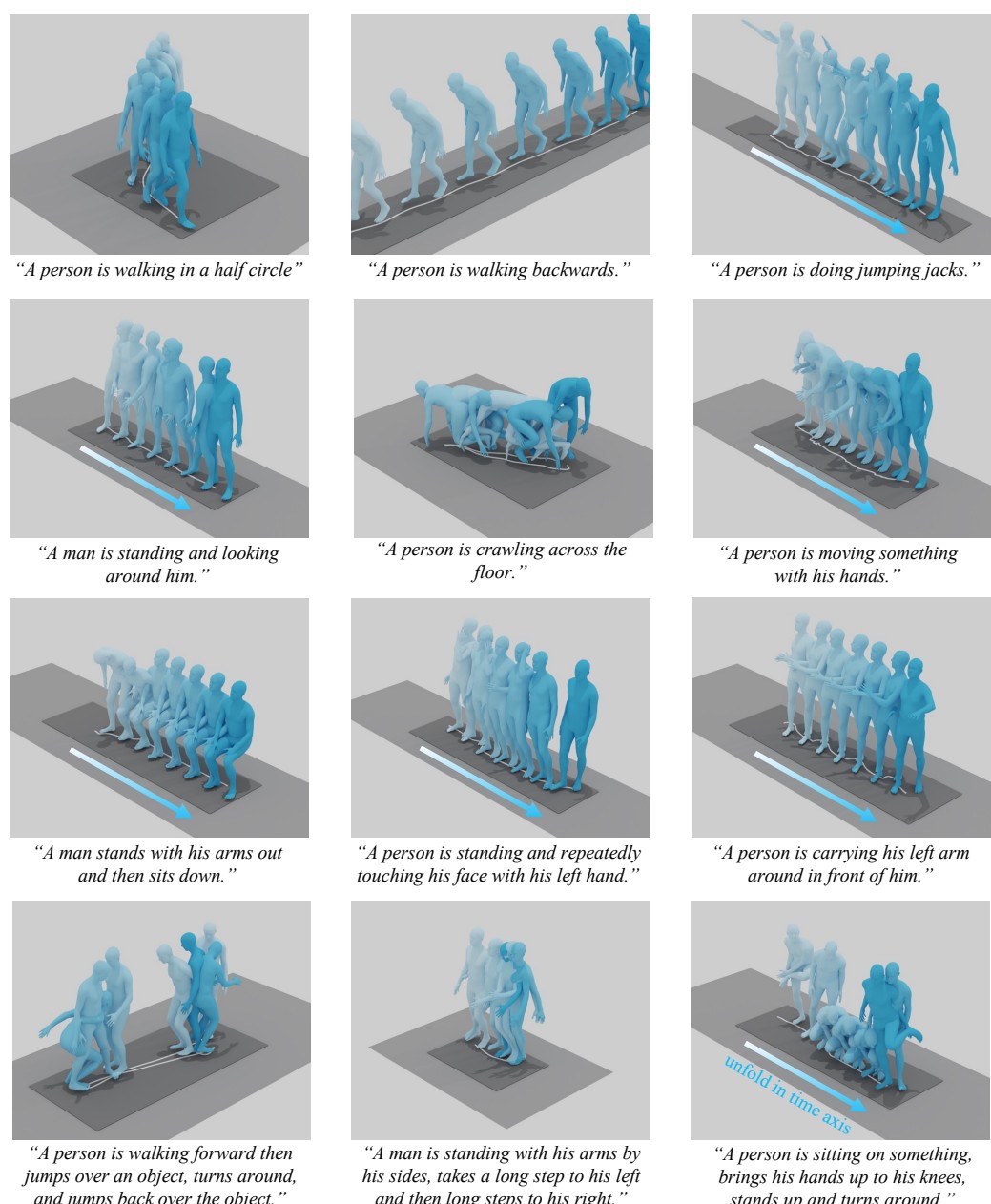

Figure A6: Qualitative results of motion-to-text captioning on HumanML3D test set.

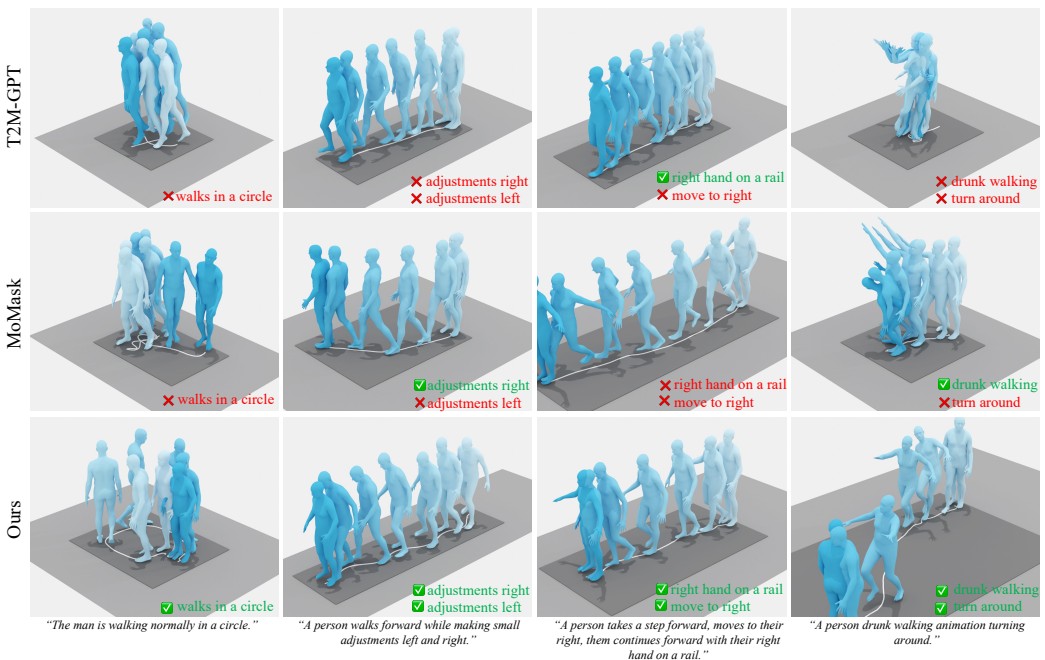

Figure A7: Qualitative results of text-to-motion generation on HumanML3D.

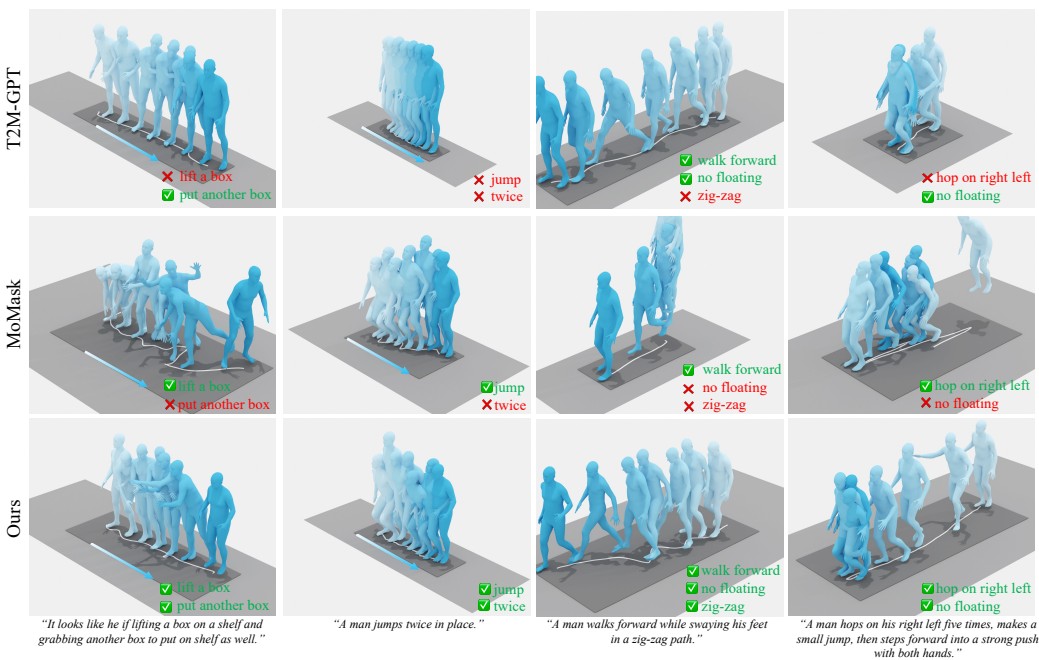

Figure A8: Qualitative results of text-to-motion generation on HumanML3D.