# OpenReview forum: "LaMP: Language-Motion Pretraining for Motion Generation, Retrieval, and Captioning"
_ICLR.cc/2025/Conference — ICLR 2025 Poster_

### Official Review · Reviewer_yD2w · 2024-10-27

**Soundness:** 3
**Presentation:** 3
**Contribution:** 3
**Rating:** 6
**Confidence:** 4

**Summary:**

The paper introduces LaMP, a novel pretraining model that aligns language and motion to enhance the performance of multiple text-motion tasks including text-to-motion generation, motion-text retrieval, and motion captioning. The authors propose a new paradigm that transitions from language-vision to language-motion latent spaces, addressing the limitations of relying on CLIP for motion-related tasks. LaMP generates motion-informative text embeddings and utilizes an autoregressive masked prediction mechanism to improve the generation process. The paper reports substantial improvements over previous methods on multiple datasets and introduces a new metric, LaMP-BertScore, for assessing the alignment of generated motions with textual descriptions.

**Strengths:**

1. The paper is well-organized and easy to follow.
2. The authors have achieved a paradigm shift from language-vision alignment to language-motion alignment. Motivated by BLIP-2, they employ four text-motion tasks for joint pretraining, leading to a more enriched latent space that encapsulates rich motion dynamics information and informative embeddings for both motion and text.
3. Through extensive experiments, the authors showcase significant improvements over state-of-the-art methods across three key tasks. Notably, their proposed method exhibits substantial performance enhancements in human motion generation tasks on the HumanML3D and KIT-ML datasets compared to previous approaches.
4. The authors introduce the LaMP-BertScore metric as a novel evaluation measure to assess the quality of generated motions in relation to textual descriptions.

**Weaknesses:**

1. Some existing studies have also emphasized the downside of aligning static images and text in CLIP, and have suggested directly aligning motion and text in the latent space, as proposed in [1]. The authors should clarify the unique distinctiveness of their method compared to existing appro aches.

2. The authors should include a computational complexity comparison between their proposed method and previous works to provide insights into resource consumption.

[1] HumanTOMATO: Text-aligned Whole-body Motion Generation

**Questions:**

None

---

> ### Author Response · Authors · 2024-11-20
> **Author Responses to Reviewer yD2w**
>
> We thank the reviewer for the valuable feedback and constructive suggestions! We hope our responses adequately address the following questions raised about our work. Please let us know if there is anything we can clarify further.
>
> **1. Difference from HumanTomato**
>
> Thanks for this insightful comment. Our method differs from HumanTomato in terms of model architecture and training objectives. While HumanTomato employs the motion encoder and text encoder from TMR [1], LaMP utilizes a transformer-based motion encoder and text encoder similar to that of CLIP. Furthermore, our training objective is not limited to contrastive loss; we designed four proxy tasks including contrastive learning, matching tasks, motion-grounded text generation, and text-grounded motion generation. This design enhances the alignment effectiveness of LaMP. As a result, we can obtain motion-aware text embeddings as well as language-aware motion embeddings, making it easier for us to finetune a motion captioning LLM. We have added the differences from previous methods in the appendix. (L831-848)
>
> **2. Inference Time**
>
> Thanks for this valuable suggestion. We have added a figure of inference time as Figure 7 in the appendix (L868). Following MoMask, we assess the efficiency and quality of motion generation compared to some methods in Table 1. The inference cost is quantified as the average inference time across 100 samples executed on a single Nvidia 2080Ti device. Our findings indicate that LaMP achieves a more advantageous balance between generation quality and computational efficiency when compared to baseline methods. This may be attributed to MoMask's multi-layer Residual-VQ structure, along with the refinement of the inference process through the Residual Transformer, resulting in reduced time consumption on our part. We have added the inference time experiment in the appendix. (L804-812)
>
> | Method         | AIT (s)   | FID ↓ |
> | -------------- | ------------- | ---------- |
> |        T2M     | 0.03      | 1.09      |
> |TM2T     | 0.76         | 1.50     |
> |MLD  | 0.22       | 0.47     |
> |MotionDiffuse   | 10.89         | 0.63    |
> |MDM    | 18.20       | 0.54     |
> |T2M-GPT    | 0.38         | 0.12      |
> |MoMask   | 0.12         | 0.04      |
> | LaMP-M2T (Ours)| 0.08        | 0.03      |
>
> Table 1. Comparisons on FID and Inference Cost.
>
> [1] TMR: Text-to-Motion Retrieval Using Contrastive 3D Human Motion Synthesis

---

### Official Review · Reviewer_ZgrR · 2024-11-01

**Soundness:** 3
**Presentation:** 3
**Contribution:** 2
**Rating:** 6
**Confidence:** 3

**Summary:**

The paper presents a language-motion pretraining approach which can address motion generation, retrieval, and caption problem. The proposed approach targets the problem of enhancing the relevancce and semantics of generated motion sequences. Experiments on multiple datasets have validated the effectiveness of the proposed approach. More specifically, the proposed approach has reasonable performance on the tasks of motion generation, retrieval, and caption.

**Strengths:**

1. The task of language-motion pretraining is of great importance to the community.
2. The paper obtains reasonable performance on several benchmarks like HumanML3D, KIT-ML datasets with different evaluation tasks.

**Weaknesses:**

1. The novelty of the paper is not clear. As LLM model is already been utilized for the motion understanding and generation, for example, MotionGPT, what are the major contribution of the paper against the previous work based LLM?

2. The algorithm is based on BERT. How about the performance if the approach is based on a large LLM, for example, LLAMA-7B?

3. For the text caption, the approach attaches another pretrained LLM model (e.g., OPT-2.7B) to generate captions. This may be due to the limitations from the BERT model. It would be better if the motion caption generation can be integrated into one model.

4. In the implementation details, what the CFG scale is set different for humanKL3D and KIT-ML datasets? (Also, what is the value for humanML3D dataset)

**Questions:**

Please address the questions raised in the weakness section.

---

> ### Author Response · Authors · 2024-11-20
> **Author Responses to Reviewer ZgrR**
>
> We thank the reviewer for the valuable feedback and constructive suggestions! We hope our responses adequately address the following questions raised about our work. Please let us know if there is anything we can clarify further.
>
> **1. Difference from previous work**
>
> Sorry for being unclear. To the best of my knowledge, there are currently two existing works titled MotionGPT, both of which focus on finetuning large language models (LLMs) for motion generation. However, a critical observation from these studies is that MotionGPT demonstrates suboptimal performance in motion generation. This limitation arises from the unresolved gap between motion and language. In contrast, we utilize a smaller model specifically designed for motion generation, employing a text encoder capable of producing motion-informative text embeddings to bridge the gap between motion and language. This approach yields significantly improved results in motion generation.
>
> Additionally, while MotionGPT requires instruction tuning of the LLM for the motion captioning task, our method for motion captioning does not necessitate such an operation, as our motion embeddings are inherently compatible with the LLM’s understanding. Consequently, we have effectively addressed the intermodal gap, and our proposed method, LaMP, can be integrated with MotionGPT. This integration would involve using LaMP to obtain motion embeddings, followed by instruction tuning of the LLM, potentially leading to enhanced performance on motion captioning tasks. We have added the differences from previous methods in the appendix. (L831-848)
>
> **2. BERT vs. LLM**
>
> Thanks for this insightful question. I believe your question revolves around why LaMP is based on BERT rather than a Large Language Model (LLM). LaMP consists of a motion transformer and a text transformer, both of which are structured based on BERT with shared parameters. If we were to use an LLM as the backbone, we would face several challenges: firstly, the number of parameters that would need fine-tuning would be substantial, resulting in significantly slower inference speeds. Our goal is to ensure that LaMP bridges the gap between motion and language, as well as facilitates motion-text retrieval; hence, a lightweight module is preferable. Otherwise, using it as a text encoder would lead to inefficiencies in processing speed.
>
> Secondly, akin to the first concern, motion embeddings cannot be directly understood by LLMs and may even compromise the LLM's inherent capabilities. LLMs are not well-suited for training with representation tasks as their objective. Consequently, the four proxy tasks we utilize for training would not be effective. Thus, we choose BERT as the backbone for this purpose. To address your concern, we also attempt to train using LLMs; however, due to their large number of parameters, they present significant challenges in terms of training feasibility.
>
> **3. Text caption model**
>
> Thanks for this constructive comment. In the field of natural language processing, decoder-only model architectures are generally considered more suitable for text generation tasks, such as GPT. BERT, as an encoder-only model, employs a bidirectional attention mask, making it more adept at text understanding and inference tasks. Thus, you are correct that BERT does indeed have its limitations. Furthermore, during the first stage of training LaMP, we also utilized motion-grounded text generation as one of the proxy tasks. To address the reviewer's concerns, we conduct an evaluation of LaMP's own motion-to-text capability, as illustrated in the first row of Table 1.
> | Method         | Top 1 ↑ |      Top 3 ↑      | MultiModal Dist ↓ | BertScore ↑ | Bleu@1 ↑ | Bleu@4 ↑ | Rouge ↑ | Cider ↑ |
> | -------------- | ------------- | ---------- | ----------------- | ----------- | -------- | -------- | ------- | ------- |
> | LaMP           | 0.397         | 0.663      | 4.076             | 17.8        | 30.1     | 6.87     | 26.6    | 16.2    |
> | LaMP-M2T (Ours)| 0.547         | 0.831      | 2.808             | 32.7        | 47.8     | 13.04    | 37.1    | 28.9    |
>
> Table 1. The quantitative results of motion captioning on the HumanML3D.
>
> **4. CFG scale**
>
> Sorry for being unclear. We did not make any further adjustments to these two parameters and simply followed MoMask. Additionally, I apologize for not specifying the CFG scale of HumanML3D in the text. In HumanML3D, CFG scale is 4, and in KIT-ML, CFG scale is 2.
> We have corrected this in the revised manuscript (L459).

---

> > ### Comment · Reviewer_ZgrR · 2024-12-03
> >
> > Thanks for the response for the original questions. The rebuttal well addressed my comments and questions. I will raise the rating to a positive side.

---

> > > ### Author Response · Authors · 2024-12-03
> > >
> > > Thank you again for your valuable suggestions and recognition!

---

> ### Author Response · Authors · 2024-11-28
> **Author Responses to Reviewer ZgrR**
>
> We thank the reviewer for the valuable feedback again! We hope our responses adequately address the questions raised about our work. Please let us know if there is anything we can clarify further.

---

### Official Review · Reviewer_S3ZK · 2024-11-03

**Soundness:** 3
**Presentation:** 3
**Contribution:** 3
**Rating:** 6
**Confidence:** 4

**Summary:**

This manuscript presents LaMP which aims at improving the alignment of language and human motion. Unlike traditional methods relying on image-text models like CLIP, LaMP directly aligns language with motion data through a dedicated language-motion latent space. This approach enhances tasks such as text-to-motion generation, motion-text retrieval, and motion captioning. LaMP leverages a transformer-based framework with four key objectives, including contrastive learning and cross-modal generation, resulting in improved performance across various datasets. The model also introduces a new metric, LaMP-BertScore, to evaluate alignment quality between text and generated motions.

**Strengths:**

1. LaMP replaces traditional image-based text embeddings with motion-focused ones, effectively improving the semantic alignment between language and motion data.
2. The model successfully tackles three critical tasks—text-to-motion generation, motion-text retrieval, and motion captioning—demonstrating versatility and effectiveness across each task.
3. The introduction of LaMP-BertScore offers a valuable metric for assessing the quality of generated motions, particularly the semantic fidelity of textual descriptions. This enhances the evaluation process in the domain of motion generation

**Weaknesses:**

1. Comparison with DLP[1] on motion-to-text generation.
2. Failure cases. Please provide some failure cases for further analysis.
3. The details about the Text encoder-decoder. How about the reconstruction metrics on the KIT and HumanML3D datasets.

[1] Cai et al. Digital Life Project: Autonomous 3D Characters with Social Intelligence. CVPR 2024.

**Questions:**

Please see the weakness part.

---

> ### Author Response · Authors · 2024-11-20
> **Author Responses to Reviewer S3ZK**
>
> We thank the reviewer for the valuable feedback and constructive suggestions! We hope our responses adequately address the following questions raised about our work. Please let us know if there is anything we can clarify further.
>
> **1. Comparison with DLP on motion-to-text generation**
>
> Thank you for your reminder! We have added the comparison with this work in the revised manuscript. Our approach fundamentally differs from DLP in that DLP utilizes a Large Language Model (LLM) as the backbone and leverages the LLM’s in-context learning capabilities for instruction tuning. In contrast, our method inputs motion into LaMP to obtain language-informative motion embeddings, facilitating a better understanding of the motion by the LLM. Additionally, there are differences in the training datasets: DLP constructed a new dataset, MoCap, and trained the LLM using this dataset, resulting in an inconsistency in our baselines. However, our approach could be integrated with DLP by extracting language-informative motion embeddings using LaMP and then applying DLP for instruction tuning to achieve improved results.
>
> Unfortunately, we find that DLP has not released their code, which prevents us from validating this quantitatively.  We will realize this after they release the code. The comparison results are shown in Table 1. We have added the brief of **DLP** in the related work. (L151)
>
> | Method            | Top 1↑ | Top 3 ↑  | MultiModal Dist ↓ | BertScore ↑ | Bleu@1 ↑ | Bleu@4 ↑ | Rouge ↑ | Cider ↑ |
> |-------------------|---------------------|---------------------|-------------------|-------------|----------|----------|---------|---------|
> | DLP | 0.551               | 0.832               | 2.813             | 35.0        | 51.1     | 15.5     | 41.9    | 36.2    |
> | LaMP-M2T (Ours)   | 0.547               | 0.831               | 2.808             | 32.7        | 47.8     | 13.04    | 37.1    | 28.9    |
>
> Table 1. The quantitative results of motion captioning on the HumanML3D, using different training datasets.
>
> **2. Failure cases**
>
> Thanks for your insightful suggestion, it can enhance the comprehensiveness and robustness of our work. We identify two primary categories of failure cases. The first category arises from issues inherent in the dataset's textual annotations. As depicted in Figure 8 (L892), the textual prompt is "A person raises his left arm." However, the dataset itself contains incorrect motion that leads to failures in our model’s outputs. Similarly, for the prompt "A person kicks with his right leg and then kicks with his left leg," the ground truth does not include the motion "kick with left leg," which results in failures in our generated results.
>
> The second one is in the context of our experiments with cartoon characters, we encountered a challenge. Cartoon characters frequently perform motions that are impossible for humans, such as flying. As illustrated in Figure 9 (L937), when we use the text prompt "A person stands and flies up," the results does not meet our expectations. This shortfall is attributed to the current dataset's relatively limited size and lack of generalizability. Moving forward, it is crucial to integrate the motions of cartoon characters with those of humans to construct a more extensive dataset to address this issue comprehensively. We have added the analysis on failure cases in the appendix. (L812-828)
>
> **3. Details about encoder-decoder and reconstruction metrics**
>
> Sorry for being unclear. There is no text encoder-decoder. For the motion encoder-decoder, we follow T2M-GPT and utilize the vanilla VQ-VAE. We employ a straightforward convolutional architecture consisting of 1D convolutions, residual blocks, and ReLU activations. We have added the framework of VQVAE as Figure 5 in the appendix (L702). Temporal downsampling is achieved via convolutions with a stride of 2, while nearest neighbor interpolation is utilized for upsampling. We also evaluate the reconstruction metrics on the HumanML3D dataset and KIT-ML dataset, as detailed in Table 2.
>
> | Dataset   | FID ↓         | MPJPE ↓  | Top 1 ↑      |
> |-----------|---------------|----------|----------------|
> | HumanML3D | 0.068 ± 0.001 | 63.06    | 0.504 ± 0.002  |
> | KIT-ML    | 0.468 ± 0.011 | 57.64    | 0.426 ± 0.005  |
> Table 2. The reconstruction results of VQVAE on HumanML3D and KIT-ML.

---

### Official Review · Reviewer_PasF · 2024-11-04

**Soundness:** 3
**Presentation:** 3
**Contribution:** 3
**Rating:** 6
**Confidence:** 4

**Summary:**

In this work, the authors point out that existing text-motion related methods largely depend on the CLIP text embeddings, yet will fall short in effectively aligning the language and motion features since CLIP is pretrained on static image-text pairs. The motion information and dynamic characteristics are overlooked. Thus, this work introduces LaMP, i.e., Language-Motion Pretraining model to generate motion-informative text embeddings, significantly enhance the relevance and semantics of generated motion sequences. With LaMP, text-to-motion generation, motion-text retrieval and motion captioning are advanced through aligned language-motion representation learning.

**Strengths:**

1. This work points out a significant problem for the current motion generative tasks, where the CLIP text embeddings are insufficient for motion generation;
2. The authors have conducted extensive experiments on text-to-motion generation, motion-text retrieval and motion captioning and show substantial improvements over baseline methods;
3. The paper writing is good and easy to follow.

**Weaknesses:**

1. CLIP is trained on large-scale image-text pair data, thus it can serve as a strong text feature extractor with strong generalization ability. However, it seems that LaMP is pretrained on a small text-motion dataset, thus the generalization ability of the LaMP features may be worse than CLIP’s. This ablation study should be provided to further verify the robustness and effectiveness of LaMP.
2. LaMP-BertScore is introduced as an evaluation metric to better measure the alignment between motion and text. However, since LaMP is trained on small-scale datasets, the metric may be not reliable and convincing.
3. In current text-motion datasets such as HumanML3D and KIT-ML, the text descriptions are very simple and may include errors, such as left-right body part mistakes. How can LaMP handle these dataset problems to produce a good text-motion alignment model?
4. Some typos: L082, pertaining → pretraining, L096, 097, an LLM → a LLM.

**Questions:**

My main concern lie in that LaMP seems to be trained on small datasets, thus the generalization ability, the proposed BertScore may be not that reliable. Meanwhile, I also want to know how to deal with the simple text descriptions and errors for LaMP?

---

> ### Author Response · Authors · 2024-11-20
> **Author Responses to Reviewer PasF**
>
> We thank the reviewer for the valuable feedback and constructive suggestions! We hope our responses adequately address the following questions raised about our work. Please let us know if there is anything we can clarify further.
>
> **1. Generalization ability of LaMP**
>
> Thank you for this insightful question. Firstly, it is indeed true that CLIP demonstrates superior generalization capabilities. However, due to the inherent differences between motion and images, CLIP has not been able to effectively align text with motion. We agree that current text-motion datasets are generally quite limited, with HumanML3D being the largest dataset available at this stage. Our method could be applied whenever a larger dataset is available. We are also actively exploring constructing larger and more comprehensive text-motion datasets, and we plan to retrain a version with enhanced generalization capabilities on a larger-scale dataset in the future. We believe that although LaMP's generalization performance is relatively lower due to the small dataset size, its performance in the domain of motion surpasses that of CLIP.
>
> To address the reviewer's concern, we have conducted additional experiments to validate the generalization of LaMP. Specifically, we utilize the LaMP model pretrained on the HumanML3D dataset as a text encoder and re-train the text-to-motion task on the KIT-ML dataset. The results are presented in Table 1.
>
> | Methods     | Top 1                | Top 2                | Top 3                | FID↓              | MultiModal Dist↓       | Diversity →         |
> |-------------|----------------------|----------------------|----------------------|-------------------|------------------------|---------------------|
> | HumanML3D-LaMP | 0.423±.006 | 0.657±.005 | 0.771±.005 | 0.226±.012  | 2.768±.022 | 10.536±.098 |
> | KIT-LaMP       | **0.479±.006** | **0.691±.005** | **0.826±.005** | **0.141±.013** | **2.70±.018** | 10.929±.100|
>
> Table 1. The quantitative results of text-to-motion generation on the KIT-ML dataset with LaMP pretrained on HumanML3D (the first row) and KIT-ML (the second row).
>
> **2. LaMP-BertScore**
>
> Thank you for this constructive comment. We acknowledge that the current training dataset is relatively small. However, this is a common drawback for all evaluation metrics in this field, e.g., FID and R-precision, as they are all trained on the same small datasets (i.e., HumanML3D and KIT-ML) as ours. Therefore, the scale of the dataset is a significant issue in this field. We initially propose LaMP-BertScore with the aim of evaluating the semantic information of generated motions. We consider this to be an important metric, as a single text prompt can correspond to multiple different motions, provided that the generated motion is semantically accurate. Reviewer S3ZK and yD2W also believe that the introduction of LaMP-BertScore offers a valuable metric for assessing the quality of generated motions, particularly the semantic fidelity of textual descriptions. This enhances the evaluation process in the domain of motion generation.
>
> **3. Dataset errors**
>
> Thank you for this insightful question. The training of LaMP is based on the premise that motion and text are correctly paired in a one-to-one manner. However, we admit that we have not specially addressed this issue. We believe that the ratio
> of such errors in the dataset should be minimal and would not significantly impact the model's performance.
>
> **4. Some typos**
>
> Thank you very much for pointing out these errors! We have corrected them in the revised version.

---

> > ### Comment · Reviewer_PasF · 2024-11-24
> >
> > Thank the authors for the detailed response. I would like to introduce the following motion datasets for your reference.
> > 1. Motion-X: A Large-scale 3D Expressive Whole-body Human Motion Dataset
> > 2. Large Motion Model for Unified Multi-Modal  Motion Generation

---

> ### Author Response · Authors · 2024-11-25
> **Author Responses to Reviewer PasF**
>
> Thank you for this suggestion!
> - For Motion-X, the precise data in Motion-X (motion capture) is identical to that of HumanML3D. However, the other data in Motion-X have been extracted using video-based algorithms, which are less accurate. We experimented with these datasets but did not achieve satisfactory results.
> - For "Large Motion Model for Unified Multi-Modal Motion Generation", the text-to-motion data in this dataset comprises HumanML3D, KIT-ML, Motion-X, and BABEL. The only difference is the BABEL dataset, which uses data from AMASS and is the same as HumanML3D. Of course, we plan to train a LaMP model based on SMPL-X on MotionVerse, but the complete dataset has not yet been released. In addition, we will cite these two papers in the related work section and provide a brief overview.
>
> Thank you again!

---

### Author Response · Authors · 2024-11-20
**Responds to all reviews and AC**

We would like to express our sincere gratitude to all the reviewers for their time and their valuable feedback. We deeply appreciate their recognition of our work, such as

**Reviewer PasF:**
"significantly enhance the relevance and semantics of generated motion sequences",
"points out a significant problem for the current motion generative tasks", "substantial improvements over baseline methods".

**Reviewer S3ZK:**
"effectively improving the semantic alignment between language and motion", "successfully tackles three critical tasks, demonstrating versatility and effectiveness across each task", "LaMP-BertScore offers a valuable metric"

**Reviewer ZgrR:**
"language-motion pretraining is of great importance to the community", "Experiments on multiple datasets have validated the effectiveness"

**Reviewer yD2W:**
"novel", "have achieved a paradigm shift from language-vision alignment to language-motion alignment", "significant improvements over state-of-the-art methods", "LaMP-BertScore metric as a novel evaluation measure"



Inspired by their thoughtful comments, we have incorporated the following changes in the revision of our paper:


- We conducted three additional experiments focusing on different aspects: the generalization ability of LaMP, the inference time for the text-to-motion task, and a failure analysis. We also provided a detailed comparison with previous methods. These sections can be found in the appendix, from L796-848.

- We have updated our manuscript, including both the main paper and the appendix, with new content highlighted in yellow. Specifically, we corrected typographical errors in L82, L96, and L97. We added a reference to DLP in the related work section at L151. Furthermore, we provided the CFG scale for HumanML3D and KIT-ML at L459, and added the structure of VQVAE at L692-L695.

We hope our responses adequately address the questions raised about our work. Please let us know if there is anything else we can clarify further.

---

### Meta-Review · Area_Chair_uQe1 · 2024-12-20

**Metareview:**

Previous methods for text-motion have relied primarily on CLIP text embedding for motion generation. However, this paper points out that CLIP is insufficient for effectively adjusting language and motion because it is pre-trained on pairs of static images and text. This study introduces LaMP, a language motion pre-training model, for generating motion information text embedding, and significantly enhances the relevance and semantics of the generated motion sequences.

This paper addresses the main problems in current motion generation tasks and proposes the LaMP model. This model can improve the consistency between text and motion more effectively than conventional image-based methods. In three important tasks (text-to-motion generation, motion text retrieval, and motion captioning), LaMP demonstrates its effectiveness and versatility. By introducing a new evaluation metric called LaMP-BertScore, the quality of generated motions can be better evaluated.

The generalization ability of the LaMP model is insufficient, and the limitations of the data set are pointed out. Specifically, since LaMP is pre-trained on a small text-motion dataset, it may have a lower generalization ability than CLIP, and the evaluation metric LaMP-BertScore may also lack reliability due to the small dataset. The dataset contains incorrect motions, and the expected results may not be obtained for specific text prompts.

Judging comprehensively from the main text, the review comments, and the rebuttal from the author, the positive points outweigh the negative points, and all reviewers gave positive evaluations. The AC considers that this paper exceeds the threshold for acceptance by ICLR.

**Additional Comments On Reviewer Discussion:**

Almost all of the reviewers' concerns were adequately addressed in the discussion.
The authors made the following revisions in accordance with the reviewers' comments and discussions.

- The authors conducted three additional experiments focusing on different aspects: LaMP's generalization ability, inference time for the text-to-motion task, and failure analysis.
- The authors also conducted a detailed comparison with the previous method.
- The authors corrected typos and supplemented missing information.

---

### Decision · Program_Chairs · 2025-01-22

Accept (Poster)